# Gene family innovation, conservation and loss on the animal stem lineage

**Daniel J Richter[1,2], Parinaz Fozouni[1,3,4], Michael B Eisen[1], Nicole King[1]\***

[1]Department of Molecular and Cell Biology, Howard Hughes Medical Institute, University of California, Berkeley, Berkeley, United States; [2]Sorbonne Universités, UPMC Univ Paris 06, CNRS UMR 7144, Adaptation et Diversité en Milieu Marin, Équipe EPEP, Station Biologique de Roscoff, Roscoff, France; [3]Medical Scientist Training Program, Biomedical Sciences Graduate Program, University of California, San Francisco, San Francisco, United States; [4]Gladstone Institutes, San Francisco, United States

**Abstract** Choanoflagellates, the closest living relatives of animals, can provide unique insights into the changes in gene content that preceded the origin of animals. However, only two choanoflagellate genomes are currently available, providing poor coverage of their diversity. We sequenced transcriptomes of 19 additional choanoflagellate species to produce a comprehensive reconstruction of the gains and losses that shaped the ancestral animal gene repertoire. We identified ~1944 gene families that originated on the animal stem lineage, of which only 39 are conserved across all animals in our study. In addition, ~372 gene families previously thought to be animal-specific, including Notch, Delta, and homologs of the animal Toll-like receptor genes, instead evolved prior to the animal-choanoflagellate divergence. Our findings contribute to an increasingly detailed portrait of the gene families that defined the biology of the Urmetazoan and that may underpin core features of extant animals.
DOI: https://doi.org/10.7554/eLife.34226.001

**\*For correspondence:**
nking@berkeley.edu

**Competing interests:** The authors declare that no competing interests exist.

## Introduction

The biology of the first animal, the 'Urmetazoan,' has fascinated and confounded biologists for more than a century (*Dujardin, 1841*; *James-Clark, 1867*; *Haeckel, 1869*; *Haeckel, 1873*; *Haeckel, 1874*; *Kent, 1880*; *Leadbeater and McCready, 2000*). What features defined the biology of the Urmetazoan, and which of those features represent animal innovations? Despite the fact that the first animals originated over 600 million years ago (*Douzery et al., 2004*; *Hedges et al., 2004*; *Peterson et al., 2004*; *Narbonne, 2005*; *Knoll, 2011*), features of their genomes can be reconstructed through phylogenetically-informed comparisons among extant animals, their closest living relatives, the choanoflagellates, and other closely related lineages (*King, 2004*; *Rokas, 2008*; *Richter and King, 2013*; *Grau-Bové et al., 2017*; *Sebé-Pedrós et al., 2017*).

Although close to 1000 genomes of animals have been sequenced (*NCBI Resource Coordinators, 2017*), only two choanoflagellate genomes have been previously published (*King et al., 2008*; *Fairclough et al., 2013*). These two choanoflagellates are the strictly unicellular *Monosiga brevicollis* and the emerging model choanoflagellate *Salpingoeca rosetta*, which differentiates into a number of sexual and asexual cell types, ranging from single cells to multicellular rosette colonies (*Fairclough et al., 2010*; *Dayel et al., 2011*; *Levin and King, 2013*). The *M. brevicollis* and *S. rosetta* genomes revealed that many genes critical for animal biology, including p53, Myc, cadherins, C-type lectins, and diverse tyrosine kinases, evolved before the divergence of animals and choanoflagellates (*King et al., 2008*; *Fairclough et al., 2013*), whereas many other genes essential for animal biology, including components of the Wnt, Notch/Delta, Hedgehog, TGF-

**eLife digest** All animals, from sea sponges and reef-building corals to elephants and humans, share a single common ancestor that lived over half a billion years ago. This single-celled predecessor evolved the ability to develop into a creature made up of many cells with specialized jobs. Reconstructing the steps in this evolutionary process has been difficult because the earliest animals were soft-bodied and microscopic and did not leave behind fossils that scientists can study.

Though their bodies have since disintegrated, many of the instructions for building the first animals live on in genes that were passed on to life forms that still exist. Scientists are trying to retrace those genes back to the first animal by comparing the genomes of living animals with their closest relatives, the choanoflagellates. Choanoflagellates are single-celled, colony-forming organisms that live in waters around the world. Comparisons with choanoflagellates may help scientists identify which genes were necessary to help animals evolve and diversify into so many different species. So far, 1,000 animal and two choanoflagellate genomes have been sequenced. But the gene repertoires of most species of choanoflagellates have yet to be analyzed.

Now, Richter et al. have cataloged the genes of 19 more species of choanoflagellates. This added information allowed them to recreate the likely gene set of the first animal and to identify genetic changes that occurred during animal evolution. The analyses showed that modern animals lost about a quarter of the genes present in their last common ancestor with choanoflagellates and gained an equal number of new genes. Richter et al. identified several dozen core animal genes that were gained and subsequently preserved throughout animal evolution. Many of these are necessary so that an embryo can develop properly, but the precise roles of some core genes remain a mystery. Most other genes that emerged in the first animals have been lost in at least one living animal.

The study of Richter et al. also showed that some very important genes in animals, including genes essential for early development and genes that help the immune system detect pathogens, predate animals. These key genes trace back to animals' last common ancestor with choanoflagellates and may have evolved new roles in animals.

DOI: https://doi.org/10.7554/eLife.34226.002

β, and innate immune pathways (e.g., Toll-like receptors) have not been detected in choanoflagellates, and therefore have been considered textbook examples of animal innovations.

Nonetheless, *M. brevicollis* and *S. rosetta* are relatively closely related (*Carr et al., 2017*), leaving the bulk of choanoflagellate diversity unexplored. Moreover, both species have demonstrably experienced gene loss, as some genes conserved among animals and non-choanoflagellates are apparently missing from *M. brevicollis* and *S. rosetta*. Examples include RNAi pathway components, which are present across eukaryotes (*Shabalina and Koonin, 2008*), the cell adhesion protein β-integrin, and T-box and Runx transcription factor families, which have been detected in the filasterean *Capsaspora owczarzaki* (*Sebé-Pedrós and Ruiz-Trillo, 2010*; *Sebé-Pedrós et al., 2010*; *Sebé-Pedrós et al., 2011*; *Sebé-Pedrós et al., 2013a*; *Ferrer-Bonet and Ruiz-Trillo, 2017*). Gene loss can lead to false negatives during ancestral genome reconstruction, and the phenomenon in choanoflagellates parallels that of animals, where two species selected for early genome projects, *Drosophila melanogaster* and *Caenorhabditis elegans*, were later found to have lost numerous genes (e.g., Hedgehog and NF-κB in *C. elegans* and fibrillar collagens in both *C. elegans* and *D. melanogaster*) that are critical for animal development and otherwise conserved across animal diversity (*C. elegans Sequencing Consortium, 1998*; *Aspöck et al., 1999*; *Gilmore, 1999*; *Rubin et al., 2000*).

To counteract the impact of gene loss in *M. brevicollis* and *S. rosetta*, and gain a more complete picture of the Urmetazoan gene catalog, we analyzed the protein coding genes of 19 previously unsequenced species of choanoflagellates representing each major known lineage (*Carr et al., 2017*). By comparing their gene catalogs with those of diverse animals and other phylogenetically relevant lineages, we have greatly expanded and refined our understanding of the genomic heritage of animals. This more comprehensive data set revealed that ~372 gene families that were previously thought to be animal-specific actually evolved prior to the divergence of choanoflagellates and animals, including gene families required for animal development (e.g., Notch/Delta) and immunity (e.g., Toll-like receptors). We find that an additional ~1944 gene families evolved along the animal

stem lineage, many of which likely underpin unique aspects of animal biology. Although most of these animal-specific genes were subsequently lost from one or more species, 39 core animal-specific genes are conserved in all animals within our data set, likely because of their importance to core features of animal biology.

## Results

### The phylogenetic distribution of animal and choanoflagellate gene families

To reconstruct the genomic landscape of animal evolution, we first cataloged the protein coding potential of nineteen diverse choanoflagellate species by sequencing and assembling their transcriptomes (*Figure 1*, *Figure 1—figure supplement 1*, *Supplementary file 1*). Because most of these species were previously little-studied in the laboratory, two important stages of this project were the establishment of growth conditions optimized for each species and the development of improved protocols for isolating choanoflagellate mRNA for cDNA library construction and sequencing (*Supplementary file 1*, Materials and methods). After performing de novo transcriptome assembly and filtering for cross-contamination, we predicted a catalog of between 18,816–61,053 unique protein-coding sequences per species. [These counts likely overestimate the true numbers of underlying protein-coding genes, as they may include multiple alternative splice variants for any given gene and redundant contigs resulting from intra-species polymorphisms or sequencing artifacts (*Grabherr et al., 2011*; *Haas et al., 2013*)].

Using multiple independent metrics, we found that the new choanoflagellate transcriptomes approximate the completeness of choanoflagellate genomes for the purposes of cataloging protein-coding genes. For example, by comparing the *S. rosetta* genome with its transcriptome, we found that 93% of *S. rosetta* genes predicted from the genome were represented in its transcriptome with coverage over at least 90% of their length (*Figure 1—figure supplement 2a*). Furthermore, compared with the genomes of *M. brevicollis* and *S. rosetta*, which contain 83 and 89%, respectively, of a benchmark set of conserved eukaryotic genes [BUSCO; (*Simão et al., 2015*)], each of the new choanoflagellate transcriptomes contains between 88–96% (*Supplementary file 2*).

We also investigated the phylogenetic diversity of the choanoflagellate species we sequenced, finding it comparable to that of animals: the average phylogenetic distance between pairs of choanoflagellates was slightly larger than the phylogenetic distance between the mouse *Mus musculus* and the sponge *Amphimedon queenslandica* (*Figure 2—figure supplement 1*). Next, we used subsets of our data to reconstruct the phylogeny of choanoflagellates. We found that the positions of two species lying on long terminal branches (*Salpingoeca dolichothecata* and *Codosiga hollandica*) were poorly supported or recovered at inconsistent locations (Materials and methods). Therefore, to avoid basing our comparative genomics efforts on a potentially incorrect phylogeny, and because the focus of our study was on reconstructing large-scale patterns of gene family evolution between animals and choanoflagellates, we designed our analyses to be independent of species relationships within either group. [For display purposes only, we relied on a consensus of previously published phylogenies (*Philippe et al., 2009*; *Burki et al., 2016*; *Carr et al., 2017*)].

We next compared the choanoflagellate gene catalogs with those of diverse animals and phylogenetically relevant outgroups (*Supplementary file 3*) to identify orthologous gene families and determine the ancestry of genes present in animals (see Materials and methods for rationale underlying inferences of gene family orthology). In summary, two features that distinguish our analyses from prior reconstructions of ancestral animal gene content are (1) the additional breadth and depth provided by 19 phylogenetically diverse and newly-sequenced choanoflagellates and (2) a probabilistic and phylogenetically-informed approach designed to avoid the artificial inflation of ancestral gene content resulting from methods that rely on binary decisions for gene family presence or absence in each species while remaining independent of currently unresolved or contentious species relationships (*Figure 2—figure supplement 2*, Materials and methods).

By grouping gene families by their phylogenetic distribution on a heat map, we were able to visualize and infer their evolutionary history, as well as their presence or absence in each species analyzed (*Figure 2*, *Figure 2—figure supplement 3*, *Figure 2—figure supplement 4*, *Supplementary file 4*; *Supplementary file 5*). Several notable observations emerged from this

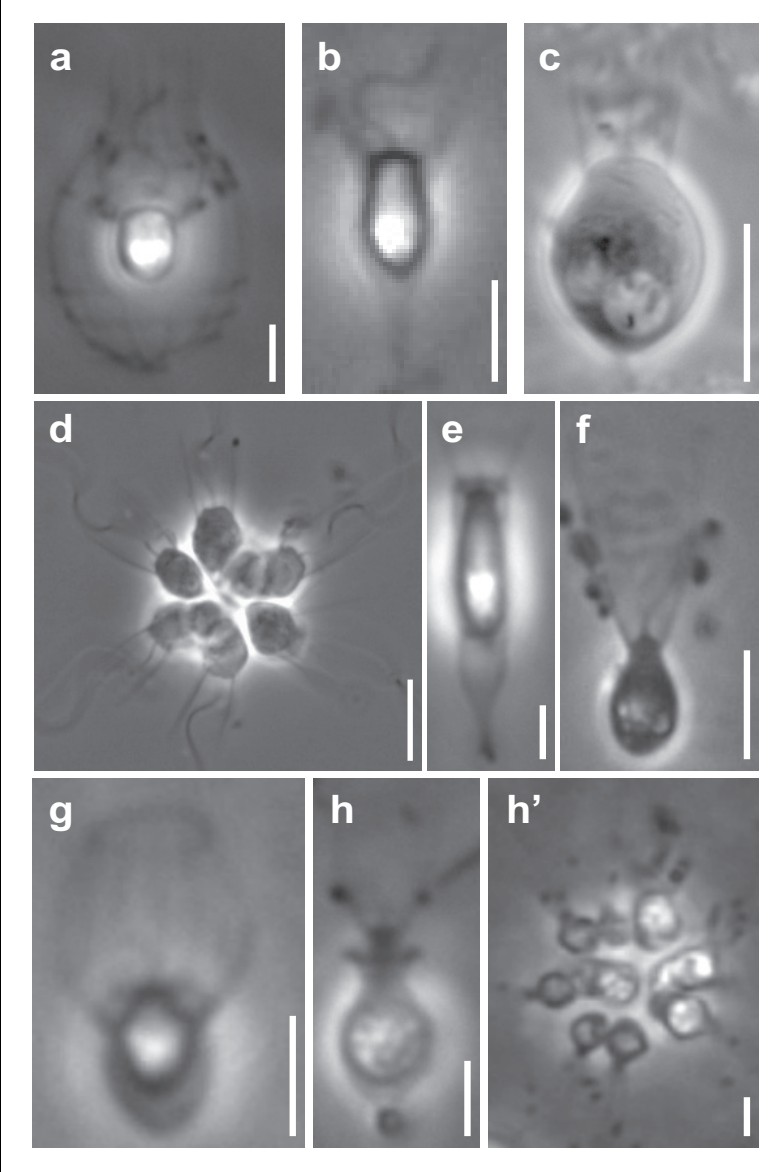

**Figure 1.** Representative choanoflagellates analyzed in this study. Choanoflagellates have diverse morphologies, including single cells, multicellular colonies, and the production in some lineages of extracellular structures. (**a**) *Diaphanoeca grandis*, within a silica-based extracellular structure called a 'lorica'. (**b**) *Acanthoeca spectabilis*, within lorica. (**c**) *Codosiga hollandica*, with a basal organic stalk. (**d**) A rosette colony of *Salpingoeca rosetta*; image courtesy of Mark Dayel. (**e**) *Salpingoeca dolichothecata*, within an organic extracellular structure called a 'theca'. (**f**) *Mylnosiga fluctuans*, with no extracellular structure. (**g**) *Didymoeca costata*, within lorica. (**h**) *Salpingoeca helianthica*, within theca. (**h'**) A rosette colony of *S. helianthica*. All scale bars represent 5 μm. Prey bacteria are visible in most panels as small black dots. Images of all choanoflagellate species sequenced in this study can be found in *Figure 1—figure supplement 1*.

DOI: https://doi.org/10.7554/eLife.34226.003

The following figure supplements are available for figure 1:

**Figure supplement 1.** Phase contrast images of species sequenced in this study.
DOI: https://doi.org/10.7554/eLife.34226.004

**Figure supplement 2.** Tests of two versus four rounds of polyA selection.
DOI: https://doi.org/10.7554/eLife.34226.005

**Figure supplement 3.** Distributions to establish thresholds within the cross-contamination removal process (**a–c**) and to eliminate noise contigs by FPKM (**d**).
DOI: https://doi.org/10.7554/eLife.34226.006

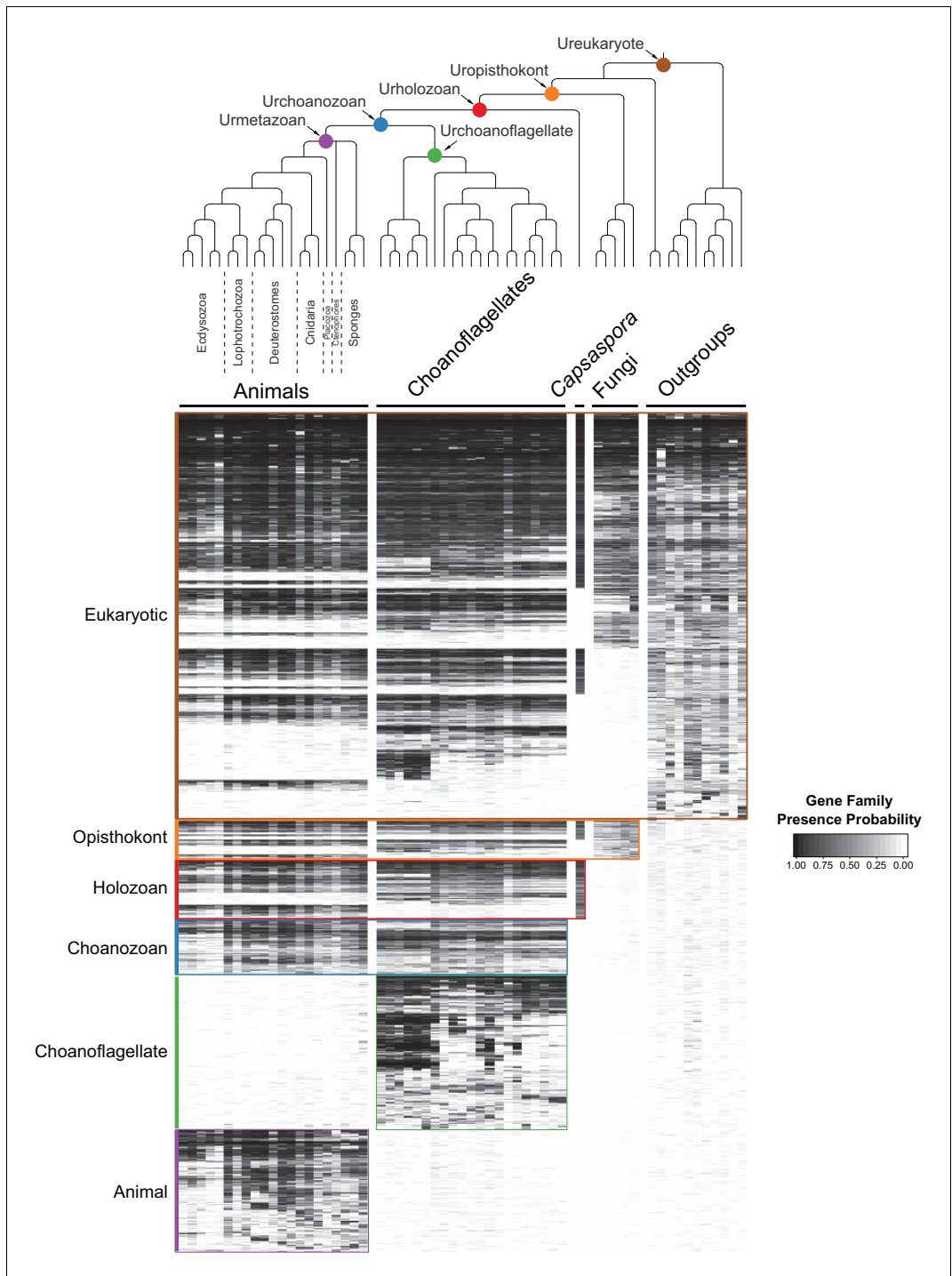

**Figure 2.** The evolution of gene families in animals, choanoflagellates and their eukaryotic relatives. Top, a consensus phylogeny (*Philippe et al., 2009*; *Burki et al., 2016*; *Carr et al., 2017*) of the species whose gene contents were compared. Each colored node represents the last common ancestor of a group of species. Bottom, a heat map of the 13,358 orthologous gene families inferred to have been present in at least one of six nodes representing common ancestors of interest: Ureukaryote, Uropisthokont, Urholozoan, Urchoanozoan, Urchoanoflagellate, and Urmetazoan (the full heat map for all gene families is shown in *Figure 2—figure supplement 4*). Each row represents a gene family. Gene families are sorted by their presence in each group of species, indicated by colored bars and boxes (eukaryotes, opisthokonts, holozoans, choanozoans, choanoflagellates and animals) and subsequently clustered within groups by uncentered Pearson correlation.

DOI: https://doi.org/10.7554/eLife.34226.007

The following source data and figure supplements are available for figure 2:

*Figure 2 continued on next page*

*Figure 2 continued*

**Source data 1.** Phylogenetic trees (in phyloXML format) used to test for contamination of choanoflagellate transcriptomes with animal sequences.
DOI: https://doi.org/10.7554/eLife.34226.020

**Figure supplement 1.** Distributions of phylogenetic diversity within choanoflagellates and within animals in our data set.
DOI: https://doi.org/10.7554/eLife.34226.008

**Figure supplement 2.** Gene family presence probabilities.
DOI: https://doi.org/10.7554/eLife.34226.009

**Figure supplement 3.** Distributions used to determine the 0.1 average probability threshold for inclusion in gene family analyses.
DOI: https://doi.org/10.7554/eLife.34226.010

**Figure supplement 4.** Full heat map of orthologous gene families present (with probability ≥0.1) in at least two species.
DOI: https://doi.org/10.7554/eLife.34226.011

**Figure supplement 5.** Heat maps with the inclusion of additional species.
DOI: https://doi.org/10.7554/eLife.34226.012

**Figure supplement 6.** Presences of selected gene families in holozoans.
DOI: https://doi.org/10.7554/eLife.34226.013

**Figure supplement 7.** Pathway components necessary for the synthesis of the essential amino acids were lost in animals.
DOI: https://doi.org/10.7554/eLife.34226.014

**Figure supplement 8.** Protein domain architectures of Notch-domain containing proteins.
DOI: https://doi.org/10.7554/eLife.34226.015

**Figure supplement 9.** The evolution of the TGF-β signaling pathway in choanozoans.
DOI: https://doi.org/10.7554/eLife.34226.016

**Figure supplement 10.** Phylogenetic tree of gene family 9066 and closely-related sequences from outside the gene family.
DOI: https://doi.org/10.7554/eLife.34226.017

**Figure supplement 11.** Distribution of gene family Pfam domain annotations within animals.
DOI: https://doi.org/10.7554/eLife.34226.018

**Figure supplement 12.** Phylogenetic trees of Notch and Delta/Serrate/Jagged.
DOI: https://doi.org/10.7554/eLife.34226.019

visualization. First, the origins of animals, choanoflagellates, and choanozoans [the monophyletic group composed of animals and choanoflagellates (*Brunet and King, 2017*)] were each accompanied by the evolution of distinct sets of gene families (i.e., synapomorphies), some of which likely underpin their unique biological features. Second, the numbers of gene families gained on the animal and choanoflagellate stem lineages are roughly equivalent (~1944 and ~2,463, respectively), indicating that the specific functions of novel gene families, not their quantity, were critical to the very different phenotypes each clade went on to have. Finally, although different sets of gene families can reliably be inferred to have been present in the last common ancestor of each group, gene family loss was rampant during animal and choanoflagellate diversification. [After these analyses were complete, several additional genomes from early-branching holozoans and animals became available. Incorporating them *post hoc* into the heat map did not substantially change any of the above observations (*Figure 2—figure supplement 5*; Materials and methods)].

## Differential retention and loss of ancestral gene families in extant animals and choanoflagellates

While the phenomenon of gene loss has been well documented in the evolution of animals and other eukaryotes (*Wolf and Koonin, 2013*; *Albalat and Cañestro, 2016*; *O'Malley et al., 2016*), it has been unclear which extant animals retained the most gene families from the Urmetazoan. Using the Urchoanozoan and Urmetazoan gene family catalogs reconstructed in this study, we ranked extant species based on their conservation of ancestral gene families (*Figure 3*, *Figure 3—figure supplement 1*). Compared with other animals in our study, the cephalochordate *Branchiostoma floridae* retains the most gene families that evolved along the animal stem lineage and also the most gene families with pre-choanozoan ancestry [extending prior observations that *B. floridae* preserved a comparatively large portion of the gene content of the last common ancestor of chordates (*Louis et al., 2012*)]. Among the non-bilaterian animal lineages, the cnidarian *Nematostella vectensis* most completely retains the Urmetazoan genetic toolkit [consistent with previous findings of conservation between *N. vectensis* and bilaterians (*Putnam et al., 2007*; *Sullivan and Finnerty, 2007*)],

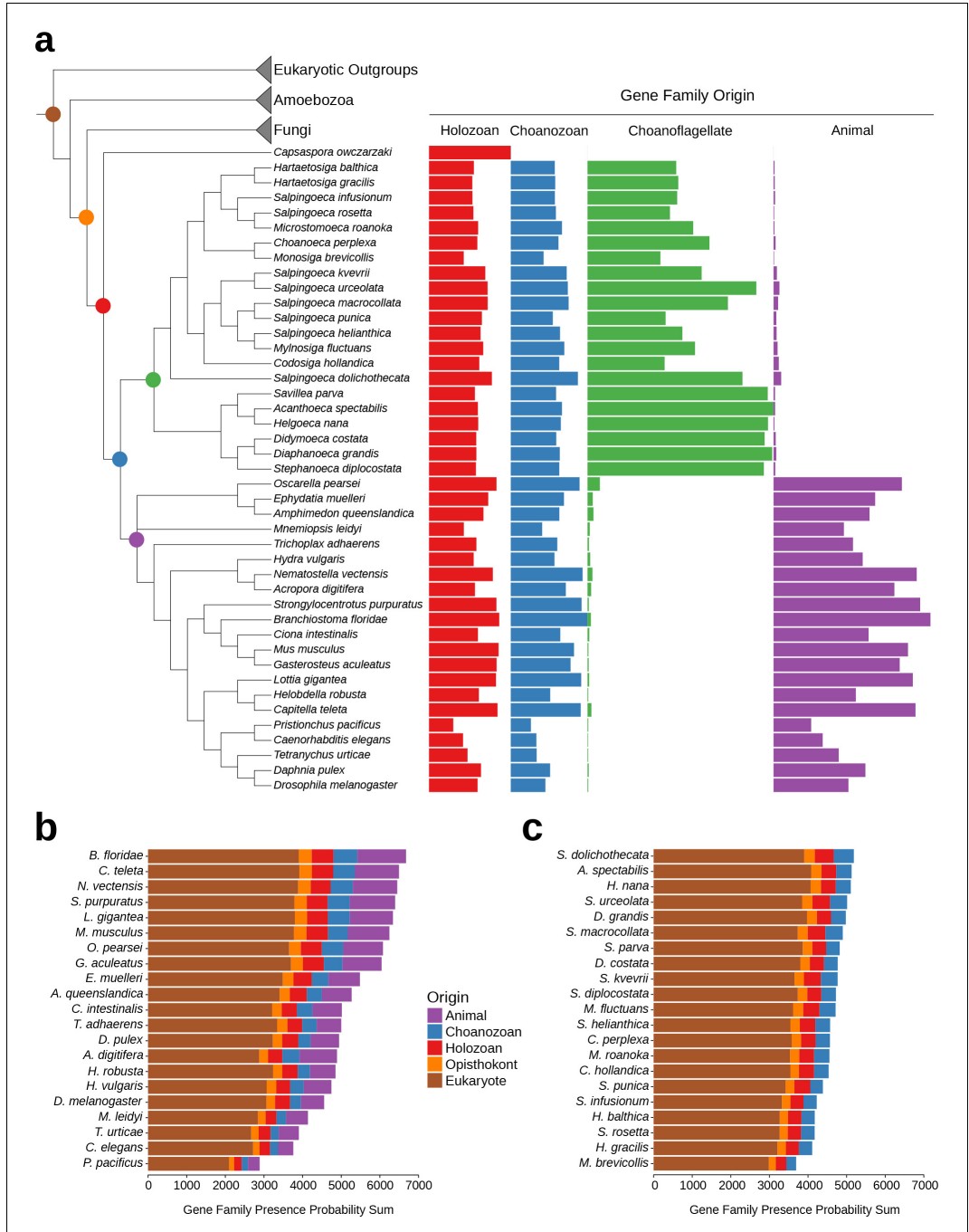

**Figure 3.** Gene family retention in animals, choanoflagellates, and *Capsaspora owczarzaki*. (a) Consensus phylogenetic tree (*Philippe et al., 2009*; *Burki et al., 2016*; *Carr et al., 2017*) with gene family retention. Gene families are divided by their origin in the last common ancestor in different groups: holozoan, choanozoan, choanoflagellate, or animal. Colors correspond to nodes indicated in the phylogenetic tree. Bars represent the sum of presence probabilities for gene families with each origin. (Note that a small sum of probabilities is assigned to certain species of choanoflagellates for animal-specific gene families, and vice versa. This is an expected result, as, in our method, every gene family is assigned a finite probability of presence in every species, producing a small background signal when summed over the approximately two thousand animal- or choanoflagellate-specific gene families. Variation in this background signal across species is due to species-specific effects on BLAST scores arising from database size and other factors, as well as intrinsically noisy scores assigned to weak BLAST hits.) (b–c) Ranked order of gene family retention for (b) animals and (c) choanoflagellates, similar to (a), but with the addition of gene families originating in the last common ancestor of Opisthokonts and of eukaryotes. Gene families originating within choanoflagellates are not included, in order to focus only on those gene families potentially shared with animals.
DOI: https://doi.org/10.7554/eLife.34226.021

The following figure supplements are available for figure 3:

*Figure 3 continued on next page*

*Figure 3 continued*

**Figure supplement 1.** Phylogenetic tree with gene family retention for all species in this study.
DOI: https://doi.org/10.7554/eLife.34226.022
**Figure supplement 2.** RNAi components in are present in choanoflagellates, but have been lost multiple times in different lineages.
DOI: https://doi.org/10.7554/eLife.34226.023
**Figure supplement 3.** The number of animal species that lost animal-specific gene families.
DOI: https://doi.org/10.7554/eLife.34226.024

followed by the sponge *Oscarella pearsei*. Importantly, *B. floridae*, *N. vectensis*, and *O. pearsei* each retain different subsets of the Urmetazoan gene catalog, as only two thirds (67%) of the genes retained in any one of these species are found in all three species. In contrast, the more rapidly evolving ecdysozoans *C. elegans*, *Pristionchus pacificus* and *Tetranychus urticae*, as well as the ctenophore *Mnemiopsis leidyi*, retain the fewest ancestral gene families, suggesting widespread gene family loss in these lineages, although the draft nature of some of their genome assemblies and high rates of sequence evolution may artificially inflate counts of missing genes.

Of the 21 choanoflagellates in our analysis, *S. dolichothecata* [which, despite the shared genus name, is not closely related to *S. rosetta* (*Carr et al., 2017*)] retains the most choanozoan-specific gene families, and therefore may be relatively more informative for comparative genomic studies of animal origins than other choanoflagellate species (*Figure 3*, *Figure 3—figure supplement 1*). Notably, the two choanoflagellate species with previously-sequenced genomes, *M. brevicollis* and *S. rosetta*, are among the choanoflagellates that have retained the fewest ancestral gene families. Thus, they are less representative of Urchoanoflagellate gene content than are most choanoflagellate species we sequenced. Indeed, several key gene families that were previously thought to be absent from choanoflagellates (due to their absence in *M. brevicollis* and *S. rosetta*) are conserved in *S. dolichothecata* and other choanoflagellates: the ancient ribonucleases Argonaute and Dicer, which are required for RNAi across eukaryotes (*Jinek and Doudna, 2009*), and holozoan gene families previously found in *C. owczarzaki* that are important for the regulation of animal development, including the transcription factors Churchill and Runx (*Sebé-Pedrós et al., 2011*) and a diagnostic domain for integrin β (*Sebé-Pedrós et al., 2010*) (*Figure 2—figure supplement 6*, *Figure 3—figure supplement 2*, *Supplementary file 6*, Materials and methods). These findings of lineage-specific gene family loss in certain animals, *M. brevicollis* and *S. rosetta* echo the more general observation that the criteria used to select species for genome sequencing also frequently select for those with streamlined genomes [e.g., (*Gu et al., 2005*)].

## Animal-specific gene families: innovation and loss

Gene families that originated on the stem lineage leading to animals are more likely to function in pathways or processes that distinguish animals from other eukaryotic groups. We identified ~1944 such animal-specific gene families (*Supplementary file 5*), including well-known developmental receptors, signaling proteins and transcription factors such as TGF-β, Hedgehog, Pax and Sox [consistent with previous reports (*Srivastava et al., 2010*; *Riesgo et al., 2014*)]. Notably, we detected many animal-specific gene families with no known function; 203 gene families (12% of total) lack any Pfam domain, and a further 50 (3%) are annotated only with Pfam domains of unknown function. The biochemical activities of these uncharacterized animal-specific gene families remain to be discovered, along with their roles in animal development and evolution.

We next sought to characterize the extent to which the ~1944 gene families that originated on the animal stem lineage were subsequently retained in the 21 animal genomes we analyzed (*Table 1*). We found only 39 gene families that are universally conserved in all 21 animal genomes in our study; we refer to these as core animal-specific gene families. This count of core animal-specific gene families is likely to be an underestimate due to methodological tradeoffs in the genome-scale analysis that we used to identify gene families (see Materials and methods). By reducing the stringency of the requirement for conservation, we identified a total of 153 gene families that were missing in no more than two animals from our data set (i.e., approximately 10%; *Figure 3—figure supplement 3*), leaving ~1791 gene families that, despite being specific to animals, were lost in three or more extant lineages. In addition, recent studies in organisms not included in our genomic data set – myxozoans, a parasitic lineage of cnidarians, and glass sponges, which develop into syncytial larvae and adults –

**Table 1.** Core animal-specific gene families that are present in all animals in this study

Representative gene names and annotations are based on a consensus from each gene family (Materials and methods). Gene families are ordered by pathway/function. *: missing in myxozoans, a lineage of parasitic cnidarians (*Chang et al., 2015*), ** missing in myxozoans and in glass sponges (*Schenkelaars et al., 2017*), two animal lineages with derived body plans.

| Gene family ID | Representative gene name(s) | Pathway/Function |
| --- | --- | --- |
| 6201 | A-kinase anchor protein 17B | gene regulation |
| 8693 | interleukin enhancer-binding factor 2 | gene regulation |
| 5720 | lethal(2) giant larvae | gene regulation |
| 5290 | mediator subunit | gene regulation |
| 6241 | mediator subunit | gene regulation |
| 7805 | MEX3 B/C | gene regulation |
| 6675 | nuclear factor 1 A/B | gene regulation |
| 3849 | T-box transcription factor TBX 2/3 | gene regulation |
| 6532 | catenin beta | Wnt |
| 4891 | catenin delta | Wnt |
| 6254 | dishevelled 1–3 | Wnt** |
| 3570 | frizzled 1/2/5/7/8 | Wnt* |
| 441 | low-density lipoprotein receptor (LRP) 1/2/4/5/6 | Wnt* |
| 5637 | transcription factor 7 (TCF/LEF) | Wnt |
| 6000 | transcription factor COE 1–4 | Wnt |
| 6831 | fermitin 1–3 | cell-cell adhesion |
| 804 | hemicentin/obscurin/titin | cell-cell adhesion |
| 4442 | integrin alpha 2/5/8 | cell-cell adhesion |
| 7164 | laminin gamma 1–3 | cell-cell adhesion |
| 8024 | vinculin | cell-cell adhesion |
| 4993 | calcium-dependent secretion activator | synapse/vesicle |
| 476 | metabotropic glutamate receptor 1–8 | synapse/vesicle |
| 7929 | receptor-type tyrosine-protein phosphatase-like N | synapse/vesicle |
| 8406 | kinase suppressor of Ras 2 | MAPK/JNK |
| 7174 | MAPK 7 | MAPK/JNK |
| 6010 | MAPK 8–10 | MAPK/JNK |
| 495 | disintegrin and metalloprotease (ADAM) | metalloprotease |
| 4051 | tetraspanin 5/14/17/33 | metalloprotease |
| 2737 | caspase 3/7/9 | apoptosis |
| 5675 | calumenin | calcium ion |
| 6842 | cyclin T 1/2 | cell cycle |
| 621 | dystonin/desmoplakin/plectin | cytoskeleton |
| 7916 | phosphorylase b kinase | glycan |
| 6512 | heparan-sulfate 6-O-sulfotransferase 1–3 | heparan sulfate |
| 7146 | inositol monophosphatase 3 | inositol |
| 6163 | protein kinase C iota/zeta | PI3K |
| 6251 | small G protein signaling modulator 1–2 | Rab GTP |
| 6366 | MAP kinase-activating death domain protein 9–11 | TNF |
| 8587 | BTB/POZ domain-containing protein 9 | ubiquitin |

DOI: https://doi.org/10.7554/eLife.34226.025

indicate that even among the 39 core animal-specific genes, some appear to be dispensable in animals with dramatically derived body plans (*Chang et al., 2015*; *Schenkelaars et al., 2017*).

Focusing on the 39 core animal-specific gene families, we asked whether they might participate in pathways known to be critical for animal biology. Indeed, this set of genes includes seven from the Wnt pathway (including Frizzled, Dishevelled, TCF/LEF and β-catenin), five involved in cell-cell adhesion (including integrin α, laminin, and vinculin), and other well-known animal gene families such as JNK, caspases, and metabotropic glutamate receptors. The 39 core animal gene families also include several that are less well characterized or whose specific contributions to animal origins and animal biology are not immediately obvious, such as two subunits of the transcription-regulating Mediator complex (*Malik and Roeder, 2010*) and the ubiquitination-associated BTB/POZ domain-containing protein 9 (*DeAndrade et al., 2012*). For comparison, choanoflagellates have a similarly small set of 75 gene families (out of ~2463 choanoflagellate-specific gene families) that are conserved in all 21 choanoflagellate species that we sampled; 27% of these gene families encode Pfam domains related to kinase signaling (including protein kinases, phosphatases and adapters; *Supplementary file 7*).

While novel features of animal biology might have evolved with the emergence of new gene families, the loss of ancient genes also influenced animal origins. We detected ~1645 gene families that evolved prior to the choanoflagellate-animal divergence that were retained in choanoflagellates and lost entirely from animals. These include gene families in pathways necessary for the biosynthesis of the amino acids leucine, isoleucine, valine, methionine, histidine, lysine and threonine (*Figure 2—figure supplement 7*, *Supplementary file 8*). The shikimic acid pathway, which is required for the synthesis of the aromatic amino acids tryptophan and phenylalanine, and other aromatic compounds, was also lost along the animal stem lineage [although subsequently regained in cnidarians through horizontal gene transfer from bacteria (*Fitzgerald and Szmant, 1997*; *Starcevic et al., 2008*)]. We thus demonstrate that components of the biosynthesis pathways for nine amino acids that are essential in animals (*Payne and Loomis, 2006*; *Guedes et al., 2011*) were lost on the animal stem lineage, and not prior to the divergence of choanoflagellates and animals. The SLN1 two-component osmosensing system, which has been shown in fungi to regulate acclimation to changes in environmental salinity (*Posas et al., 1996*), is also conserved in choanoflagellates but absent in animals. [Although these amino acid synthesis and osmosensing pathway components were retained in choanoflagellates, several other gene families involved in diverse biosynthetic pathways were instead lost on the choanoflagellate stem lineage (*Supplementary file 9*)]. Together, the ensemble of animal gene family losses reflects the substantial changes in metabolism and ecology that likely occurred during early animal evolution.

## Choanozoan-specific gene families: innovation and loss

In addition to the set of gene families that evolved on the animal stem lineage, those that originated on the holozoan and choanozoan stem lineages also contributed to the genomic heritage of animals. Our increased sampling of choanoflagellate diversity allowed us to ask whether gene families previously thought to have been animal innovations, due to their absence from *M. brevicollis*, *S. rosetta* and other outgroups, may in fact have evolved before the divergence of animals and choanoflagellates. We found that ~372 gene families previously thought to be restricted to animals have homologs in one or more of the 19 newly sequenced choanoflagellates (*Supplementary file 5*; see *Supplementary file 10* for a list of pathways with components gained or lost on the Choanozoan stem lineage).

Within this set of genes are the Notch receptor and its ligand Delta/Serrate/Jagged (hereafter Delta), which interact to regulate proliferation, differentiation and apoptosis during animal developmental patterning (*Artavanis-Tsakonas et al., 1999*). Intact homologs of these important signaling proteins have never previously been detected in non-animals, although some of their constituent protein domains were previously found in *M. brevicollis*, *S. rosetta* and *C. owczarzaki* (*King et al., 2008*; *Suga et al., 2013*). In our expanded choanoflagellate data set, we detected a clear Notch homolog in *Mylnosiga fluctuans* with the prototypical EGF (epidermal growth factor), Notch, transmembrane and Ank (ankyrin) domains in the canonical order, while five other choanoflagellates contain a subset of the typical protein domains of Notch proteins (*Figure 2—figure supplement 6*, *Figure 2—figure supplement 8*). Similarly, the choanoflagellate species *S. dolichothecata* expresses a protein containing both of the diagnostic domains of animal Delta (MNNL and DSL, both of which

were previously thought to be animal-specific) and four other choanoflagellate species express one of the two diagnostic domains of Delta, but not both. The distributions of Notch and Delta in choanoflagellates suggest that they were present in the Urchoanozoan and subsequently lost from most (but not all) choanoflagellates, although it is formally possible that they evolved convergently through shuffling of the same protein domains in animals and in choanoflagellates. A similar portrait emerges for the cadherins Flamingo and Protocadherin (*Chae et al., 1999*; *Usui et al., 1999*; *Frank and Kemler, 2002*) that were previously thought to be animal-specific, but are found in a subset of choanoflagellates within our data set (*Figure 2—figure supplement 6*, Materials and methods).

We also found evidence that numerous gene families and pathways that originated in animals arose through shuffling of more ancient protein domains and genes that were already present in the Urchoanozoan (*Ekman et al., 2007*; *King et al., 2008*; *Grau-Bové et al., 2017*). For example, the new choanoflagellate gene catalogs confirm that several signature animal signaling pathways, such as Hedgehog, Wnt, JNK, JAK-STAT, and Hippo, are composed of a mixture of gene families that were present in the Urchoanozoan and others that evolved later on the animal stem lineage or within animals (*Snell et al., 2006*; *Adamska et al., 2007*; *Hausmann et al., 2009*; *Richards and Degnan, 2009*; *Srivastava et al., 2010*; *Sebé-Pedrós et al., 2012*; *Babonis and Martindale, 2017*) (*Supplementary file 8*). For another animal signaling pathway, TGF-β, the critical ligand, receptor and transcription factor gene families are composed of animal-specific domain architectures (*Heldin et al., 1997*; *Munger et al., 1997*), although all three contain constituent protein domains that evolved on the choanozoan stem lineage (*Figure 2—figure supplement 9*).

## The pre-animal origins of the animal innate immunity pathway

One surprise from our analyses was the existence in choanoflagellates of genes required for innate immunity in animals. Although innate immunity is a feature of both animal and plant biology, the receptors and pathways used by these evolutionarily distant organisms are thought to have evolved independently (*Ausubel, 2005*). The animal immune response is initiated, in part, when potential pathogens stimulate (either directly or indirectly) the Toll-like receptors (TLRs), which have previously been detected only in animals (*Leulier and Lemaitre, 2008*). Importantly, although TLRs are found in nearly all bilaterians and cnidarians, they are absent from placozoans, ctenophores, and sponges [proteins with similar, but incomplete domain architectures have been detected in sponges (*Miller et al., 2007*; *Riesgo et al., 2014*)] and were therefore thought to have evolved after the origin of animals.

We found that 14 of 21 sequenced choanoflagellates encode clear homologs of animal TLRs (*Figure 4a,c*), implying that TLRs first evolved on the Urchoanozoan stem lineage (Materials and methods). Animal and choanoflagellate TLRs are composed of an N-terminal signal peptide, multiple leucine rich repeat (LRR) domains, a transmembrane domain, and an intracellular Toll/interleukin-1 receptor/resistance (TIR) domain. In the canonical TLR signaling pathway, the interaction of the intracellular TIR domain of Toll-like receptors with TIR domains on adapter proteins (e. g., MyD88) initiates one of a number of potential downstream signaling cascades that ultimately lead to activation of the NF-κB transcription factor (*Janeway and Medzhitov, 2002*).

To investigate whether the Urchoanozoan TLR might have activated a downstream signaling pathway that resembled the canonical TLR pathway in animals, we searched for TLR adapters, downstream kinases and NF-κB in choanoflagellates (*Figure 4c*, *Supplementary file 6*). While many choanoflagellates encode NF-κB, we found no evidence for two critical Death domain-containing proteins involved in the TLR-dependent activation of NF-κB: the adapter protein MyD88 (*Wiens et al., 2005*; *Gauthier et al., 2010*) and the downstream kinase IRAK (*Song et al., 2012*). However, we did detect two new classes of choanoflagellate-specific proteins that pair kinase domains directly with LRR and/or TIR domains, potentially bypassing the need to recruit kinases into multi-protein signaling complexes (*Figure 4b*): TLR-like proteins with an intracellular kinase domain positioned between the transmembrane domain and TIR domain (which we provisionally term 'Kinase TLRs') and proteins encoding TIR and kinase domains, but lacking a transmembrane domain (which we provisionally term 'Kinase TIRs'). In addition, we detected homologs of the TIR-containing adapter SARM1, a multi-functional protein that can trigger both NF-κB-dependent and NF-κB-independent responses (*Couillault et al., 2004*; *Sethman and Hawiger, 2013*; *Liu et al., 2014*). Choanoflagellate SARM1 homologs contain a conserved glutamic acid residue that is necessary for SARM1

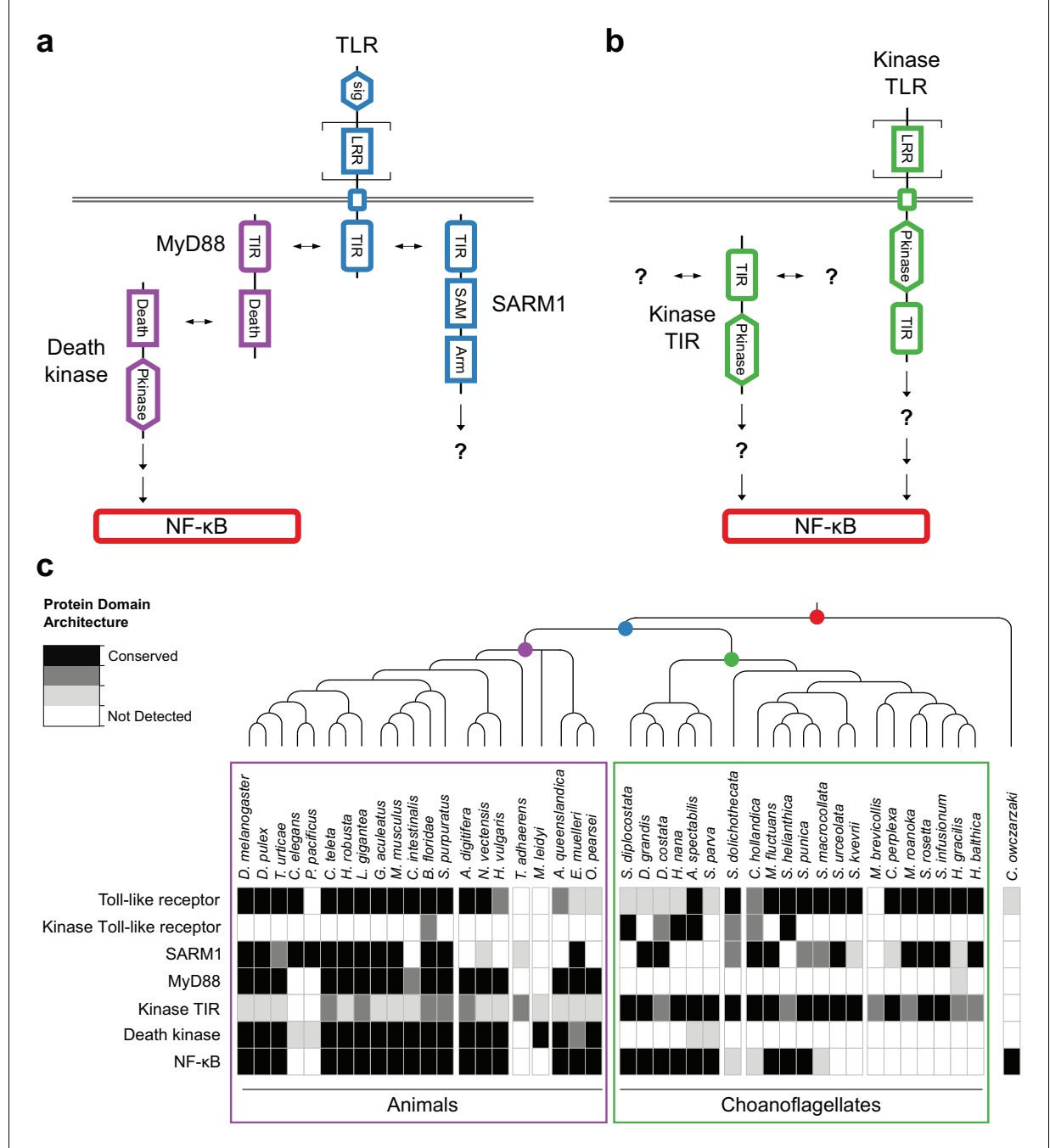

**Figure 4.** Evolution of the TLR signaling pathway. Components of the canonical TLR pathway (**a**) and a potential choanoflagellate Kinase TLR signaling pathway (**b**), with their canonical domain architectures and colored by their inferred ancestral origin (blue = choanozoan ancestry, purple = metazoan ancestry, green = choanoflagellate ancestry, and red = holozoan ancestry). Question marks denote steps of the signaling pathway and/or interaction partners that are hypothesized, but untested. (**c**) Presence of receptors, adapters, kinases and the transcription factor NF-κB in animals, choanoflagellates and *C. owczarzaki*.

DOI: https://doi.org/10.7554/eLife.34226.026

The following figure supplements are available for figure 4:

**Figure supplement 1.** Alignment of gene family 6840, which contains animal SARM1 proteins.

DOI: https://doi.org/10.7554/eLife.34226.027

**Figure supplement 2.** Phylogenetic tree of proteins in our data set containing any of the following pairs of Pfam protein domains: LRR and kinase, kinase and TIR, or LRR and TIR.

DOI: https://doi.org/10.7554/eLife.34226.028

NADase activity in animals (*Essuman et al., 2017*) (*Figure 4—figure supplement 1*). Finally, although we did not detect most animal cytosolic innate immune sensors in choanoflagellates, including the LRR-containing NLR family, ALRs, MAVS, MDA5 and RIG-I, we did find evidence for both cGAS and STING in diverse choanoflagellates [as previously reported in *M. brevicollis* (*Wu et al., 2014*)]. Thus, critical components of the animal innate immune pathway, including both extracellular and intracellular pattern sensing receptors, predate animal origins.

## Discussion

Our increased sampling of choanoflagellates reveals how the Urmetazoan genome evolved as a mosaic of old, new, rearranged, and repurposed protein domains, genes and pathways. We have identified ~8418 gene families that were present in the Urmetazoan [consistent with recent findings (*Simakov and Kawashima, 2017*)], about 75% of which were also present in the Urchoanozoan and the remainder of which evolved on the animal stem lineage (*Supplementary file 5*). The patchwork ancestry of the Urmetazoan genome is illustrated by the fact that many gene families responsible for animal development, immunity and multicellular organization evolved through shuffling of protein domains that first originated in the choanozoan stem lineage together with ancient or animal-specific domains (e.g. the TGF-β ligand and receptor; *Figure 2—figure supplement 6*, *Figure 2—figure supplement 9*). In addition, other gene families found in the Urchoanozoan were subsequently combined into new pathways in the animal stem lineage along with newly evolved genes (e.g., the TLR and Hedgehog pathways; *Figure 4*, *Supplementary file 8*). Moreover, the history of the Urmetazoan genome is not simply one of innovation and co-option, as ~1,645 Urchoanozoan genes were lost on the animal stem lineage, including genes for the synthesis of nine essential amino acids *[Figure 2—figure supplement 7*, *Supplementary file 8*; (*Payne and Loomis, 2006*; *Guedes et al., 2011*; *Erives and Fassler, 2015*)]. A study based on similar methodology that incorporated two of the 21 choanoflagellate species analyzed here (*M. brevicollis* and *S. rosetta*) was recently published and found a similar pattern of gene innovation on the animal stem lineage, while also identifying many of the same core animal-specific genes (*Paps and Holland, 2018*).

The origin of multicellularity in animals provided novel niches for bacteria to exploit, requiring the first animals to evolve new mechanisms for mediating interactions with pathogenic and commensal bacteria. In addition, the progenitors of animals interacted with bacteria – both as prey and pathogens – and the roots of animal innate immunity clearly predate animal origins. We have found that choanoflagellates express TLRs, transmembrane receptors that trigger the animal innate immune response, as well as its canonical downstream signaling target, NF-κB, suggesting that both existed in the Urchoanozoan (*Figure 4a,c*). Like modern choanoflagellates and sponges, the Urchoanozoan likely preyed upon bacteria (*McFall-Ngai et al., 2013*; *Richter and King, 2013*), and bacterial cues can induce life history transitions in choanoflagellates (*Alegado et al., 2012*; *Woznica et al., 2016*; *2017*), although the mechanisms by which choanoflagellates capture bacteria and sense bacterial cues are unknown. We hypothesize that the core TLR/NF-κB pathway functioned in prey sensing, immunity, or more complex processes in the Urchoanozoan that subsequently formed the basis of a self-defense system in animals. Because critical pathway members linking TLR and NF-κB appear to be animal innovations (e.g., MyD88), the animal signaling pathway may have evolved to diversify downstream signaling processes to tailor responses in a multicellular context. This pathway diversification may have included the evolution of roles in development, as TLRs have been implicated in both NF-κB-dependent and NF-κB-independent developmental signaling (in addition to their functions in immunity) in bilaterians and in the cnidarian *N. vectensis* (*Leulier and Lemaitre, 2008*; *Brennan et al., 2017*). The uncharacterized choanoflagellate-specific Kinase TLRs and Kinase TIRs (*Figure 4b,c*) may function as part of a streamlined signaling pathway that mediates responses to extracellular cues, including bacteria, although further research will be required to test this hypothesis.

Our study provides a detailed view of the changes in gene content that laid the foundation for animal origins. Innovations in gene and protein regulation in the Urmetazoan also likely contributed to animal evolution, as features of animal phosphoproteome remodeling, gene co-regulation and alternative splicing (but not animal promoter types and enhancers) have been found in *C. owczarzaki* (*Sebé-Pedrós et al., 2013b*; *2016a*; *2016b*) and in the holozoan *Creolimax fragrantissima* (*de Mendoza et al., 2015*). Multicellularity has also evolved independently in a number of other

eukaryotic lineages, including slime molds, brown algae, fungi, and chlorophytes. In each transition to multicellularity, the underlying genomic changes are distinct from those that occurred on the animal stem lineage. For example, in the social amoeba *Dictyostelium discoideum*, many novel gene families are involved in extracellular sensing (*Glöckner et al., 2016*), similar to the marked increase in signal transduction gene families found in the multicellular brown alga *Ectocarpus siliculosus* (*Cock et al., 2010*). Gene innovations in multicellular ascomycete fungi are enriched for functions related to endomembrane organelles (*Nguyen et al., 2017*) and the gene complement of the multicellular green alga *Volvox carteri* is largely distinguished from its unicellular relative *Chlamydomonas reinhardtii* by expansions or contractions within gene families, rather than the evolution of new families (*Prochnik et al., 2010*). Through our analyses of genomes and transcriptomes representing the full breadth of choanoflagellate and animal diversity, we have provided a genome-scale overview of the gene families whose invention or co-option distinguished the Urmetazoan from all other organisms and therefore may have provided a basis for the evolution of unique mechanisms regulating development, homeostasis and immunity in animals.

## Materials and methods

The sections on Quality trimming, Error correction, *De novo* transcriptome assembly, Identification and removal of cross-contamination, Prediction of amino acid sequences from assembled transcripts and elimination of redundant transcripts, and Measurement of expression levels and elimination of noise transcripts were described in (*Peña et al., 2016*). They are repeated here for convenience and clarity (with modifications to the text but not to the underlying methods).

### Origin of cultures

We acquired 18 of 20 cultures used for transcriptome sequencing from external sources (*Supplementary file 1*). We isolated the remaining two cultures, *Acanthoeca spectabilis* (Virginia) ATCC PRA-387 and *Codosiga hollandica* ATCC PRA-388. (*A. spectabilis* (Virginia) is a different isolate from *A. spectabilis* ATCC PRA-103, which was originally collected in Australia.) We collected the water sample from which *A. spectabilis* (Virginia) was isolated on December 19, 2007 near Hog Island, Virginia (GPS coordinates: 37.472502,–75.816018) and we isolated *C. hollandica* from a sample collected on June 25, 2008 from Madeira, Portugal (GPS coordinates: 32.762222,–17.125833). *C. hollandica* was formally described in *Carr et al. (2017)*.

We isolated choanoflagellates with a micromanipulator and a manual microinjector (PatchMan NP 2 and CellTram vario 5176 (Eppendorf, Hamburg, Germany) for *A. spectabilis*, and XenoWorks Micromanipulator and Analog Microinjector (Sutter Instrument, Novato, California, United States) for *C. hollandica*). We pulled glass needles used for isolation from 1 mm diameter borosilicate glass (GB100-10, Science Products GmbH, Hofheim, Germany) using a Flaming/Brown needle puller (P-87, Sutter Instrument) with the following program: heat = 820, pull = 50, velocity = 140, time = 44. We polished and sterilized needles by passing them briefly through a low flame. We used a separate needle for each attempted isolation, transferring single cells into separate culture flasks containing appropriate growth medium (see *Supplementary file 1*). In order to reduce the possibility of contamination during the isolation procedure, we generated sterile air flow across the microscope and micromanipulator apparatus using a HEPA-type air purifier (HAP412BN, Holmes, Boca Raton, Florida, United States).

One culture we obtained from ATCC, *Salpingoeca infusionum*, was contaminated by an unidentified biflagellated unicellular eukaryote. To remove the contaminant from the culture, we counted then diluted cells into separate wells of two 24-well plates. After 7 days of growth, we found 4 of 48 wells containing only *S. infusionum* and bacteria, one well containing only the contaminant and bacteria, and the remaining 43 wells containing only bacteria. We selected one of the four wells containing only *S. infusionum* for use in transcriptome sequencing.

### Antibiotic treatment and optimization of culture conditions

Choanoflagellates are co-isolated with diverse bacterial prey, which serve as a food source. In order to limit bacterial diversity to the species that led to optimal choanoflagellate growth, we treated each culture with a panel of ten different antibiotics (*Supplementary file 11*). We obtained all antibiotics from Thermo Fisher Scientific (Waltham, Massachusetts, United States) with the exception of

erythromycin, gentamicin, ofloxacin, and polymyxin B (Sigma-Aldrich, St. Louis, Missouri, United States). We sterilized antibiotic solutions before use by filtration through a 0.22 µm syringe filter (Thermo Fisher Scientific) in a laminar flow hood. We initially treated each culture with all ten antibiotics. We selected initial treatments that decreased bacterial density and diversity, and then repeatedly diluted the cultures into fresh medium with the same antibiotic until no further change in bacterial density or diversity was observed. We then re-treated each of these cultures with an additional test of all ten antibiotics, as their modified bacterial communities often responded differently from their initial communities. We repeated successive rounds of treatment until no further reduction in bacterial density or diversity was observed (*Supplementary file 1*).

We tested a range of concentrations of different temperatures and growth media (*Supplementary file 12*) in order to maximize choanoflagellate cell density, with three types of nutrient sources: quinoa grains, proteose peptone/yeast extract, and cereal grass. We used filtered water (Milli-Q, Millipore, Burlington, Massachusetts) when preparing all solutions. For marine species, we used 32.9 grams per liter of artificial seawater (Tropic Marin, Montague, Massachusetts, United States). We used proteose peptone (Sigma-Aldrich Chemical) at a final concentration of 0.002% (w/v) and yeast extract (Becton Dickinson Biosciences, San Jose, California, United States) at a final concentration of 0.0004% (w/v). To prepare cereal grass media (also known as chlorophyll alfalfa, Basic Science Supplies, St. Augustine, Florida, United States), we added it to autoclaved water and allowed it to steep until cool. Once cool, we removed large particles of cereal grass by repeated vacuum filtration through a ceramic Buchner funnel with a double layer of Grade one cellulose filter paper (Whatman, GE Healthcare Life Sciences, Marlborough, Massachusetts, United States). We autoclaved organic quinoa grains and added them to the medium after filtration, with roughly two grains added per 25 cm$^2$ of culture vessel surface area. We measured final nutrient content of each type of medium by Flow Injection Analysis at the University of California, Santa Barbara Marine Science Institute (*Supplementary file 3*). We tested buffered medium for two freshwater species that experienced lowered pH during growth, *C. hollandica* (pH 5.5) and *Salpingoeca punica* (pH 5), using 50 mM HEPES (Thermo Fisher Scientific) adjusted to a pH of 7. We sterilized all media with a 0.22 µm vacuum filter (Steritop, Millipore) in a laminar flow hood prior to use.

We selected final culture conditions that maximized choanoflagellate density and variety of cell types present, as we hypothesized that different cell types, each potentially expressing different subsets of genes, would yield the greatest diversity of transcripts for sequencing. We defined five generic choanoflagellate cell types: 'attached': attached to the bottom of the culture vessel or to a piece of floating debris, either directly, within a theca, within a lorica, or on a stalk; 'slow swimmer': a typical swimming cell; 'fast swimmer': a cell with reduced collar length swimming at higher speed; 'free-swimming colonial': in a colony swimming in the water column; 'attached colonial': in a colony attached to a stalk; 'passively suspended': suspended in the water column, either naked or within a lorica. See (*Dayel et al., 2011*; *Carr et al., 2017*) for further information on choanoflagellate cell types and (*Leadbeater, 2015*; *Richter and Nitsche, 2016*) for descriptions of extracellular structures (thecae, loricae, etc.).

## Growth of cultures in large batches in preparation for RNA isolation

We routinely grew choanoflagellates in 25 cm$^2$ angled neck cell culture flasks with 0.2 µm vented caps (Corning Life Sciences, Corning, New York, United States) containing 25 ml of medium. We performed all cell culture work in a laminar flow hood. To reduce the possibility of cross-contamination among samples in the hood, we dispensed media into growth vessels prior to the introduction of cultures, we only worked with a single culture at a time, and we cleaned thoroughly with 70% ethanol before and after introducing cultures. To grow and collect large amounts of cells for RNA preparation, we used different growth vessels, volumes, growth durations and centrifugation times as appropriate to each culture (*Supplementary file 1*). Vessels included long neck Pyrex glass culture flasks (Corning), 150 mm plastic tissue culture dishes (Becton Dickinson), and 75 cm$^2$ angled neck cell culture flasks with 0.2 µm vented caps (Corning).

We harvested cultures depending on the cell types present (*Supplementary file 1*). For cultures with five percent or fewer attached cells, we collected liquid by pipetting without scraping (to reduce the number of bacteria collected). For cultures containing between 5 and 90 percent attached cells, we harvested single plates by pipetting after scraping cells from the bottom of the plate. For cultures with 90 percent or greater attached cells, we combined multiple plates as follows:

we removed and discarded the liquid from the first plate by pipetting, added 50 ml of either artificial sea water or filtered water, as appropriate, scraped cells from the plate, removed the liquid from the second plate, transferred the liquid from the first to the second plate, and repeated the procedure on subsequent plates. For cultures containing quinoa grains or large bacterial debris, we filtered with a 40 µm cell strainer (Fisher) after collection.

We pelleted cells in 50 ml conical tubes at 3220 x g in a refrigerated centrifuge at 4°C, removed the first 47.5 ml of supernatant by pipetting, and the last 2.5 ml by aspiration. When we harvested more than 50 ml for a culture, we spun in separate tubes, removed all but 2.5 ml of supernatant, resuspended, combined into a single 50 ml conical tube, and repeated the centrifugation as above. We flash froze pellets in liquid nitrogen and stored them at −80°C. To reduce the possibility of cross-contamination, we harvested and centrifuged each culture separately, we used disposable plastic pipette tubes, conical tubes, and cell scrapers, and we cleaned all other material (bench tops, pipettes, etc.) with ELIMINase (Decon Laboratories, King of Prussia, Pennsylvania, United States) between cultures.

## RNA isolation

We isolated total RNA from cell pellets with the RNAqueous kit (Ambion, Thermo Fisher Scientific). We modified the manufacturer's protocol to double the amount of lysis buffer, in order to increase RNA yield and decrease degradation. We performed both optional steps after adding lysis buffer: we spun for 3 min at top speed at 1°C and passed the supernatant through a 25 gauge syringe needle several times. We used the minimum suggested volumes in the two elution steps (40 and 10 µl). We measured RNA concentration using a NanoDrop ND-1000 spectrophotometer (Thermo Fisher Scientific).

For all species except *C. hollandica*, we immediately proceeded to digest genomic DNA using the TURBO DNA-free kit (Ambion, Thermo Fisher Scientific), following the manufacturer's protocol with a 30 min incubation. After digestion, we removed DNase with DNase Inactivation Reagent for all species except *S. punica*, whose RNA extract was incompatible with the reagent. We instead removed DNase from *S. punica* by extracting with two volumes of pH eight phenol:chloroform:isoamyl alcohol, removing residual phenol with two volumes of chloroform:isoamyl alcohol, and precipitating with 0.3 M sodium acetate pH 5.2 (all three from Sigma-Aldrich), 25 µg of GlycoBlue (Thermo Fisher Scientific) as a carrier and two volumes of 100% ethanol. We washed the pellet in 70% ethanol and resuspended in 50 µl of nuclease-free water. For *Didymoeca costata*, after DNase removal with the Inactivation Reagent, the RNA still appeared to be slightly contaminated with protein, so we performed a pH eight phenol:chloroform extraction to remove it. For *C. hollandica*, we observed significant total RNA degradation in the presence of DNase buffer. Instead, to remove genomic DNA we performed three successive rounds of extraction with pH 4.5 phenol:chloroform:isoamyl alcohol (Sigma-Aldrich), followed by the chloroform:isoamyl and precipitation steps described above. To reduce the possibility of cross-contamination among samples, we performed RNA isolation and DNase digestion for a single culture at a time, we used disposable materials when possible, and we cleaned all other materials (bench tops, centrifuges, dry baths, pipettes, etc.) thoroughly with ELIMINase before use.

We evaluated total RNA on Bioanalyzer 2100 RNA Pico chips (Agilent Technologies, Santa Clara, California, United States) with four criteria to be considered high-quality: (1) four distinct ribosomal RNA peaks (16S and 23S for bacteria, 18S and 28S for choanoflagellates), (2) low signal in all other regions, as a non-ribosomal signal is evidence of degradation, (3) at least a 1:1 ratio of 28S ribosomal area to 18S ribosomal area, since 28S ribosomal RNA is likely to degrade more easily than is 18S ribosomal RNA, and (4) an RNA Integrity Number (RIN) of 7 or greater (*Schroeder et al., 2006*). (We note that the Bioanalyzer software could not calculate RIN for several cultures.) If we were not able to obtain high-quality total RNA after the first attempt for any culture, we repeated cell growth, centrifugation and total RNA isolation up to a maximum of 5 times, and selected the best available total RNA sample to use for transcriptome sequencing. We produced a rough estimate of the amount of choanoflagellate total RNA present in each sample by calculating the ratio of choanoflagellate to bacterial ribosomal RNA peaks (18S vs. 16S and 28S vs. 23S) and multiplying the resulting fraction by the total amount of RNA present in the sample (*Supplementary file 1*).

## Test of polyA selection to separate choanoflagellate from bacterial RNA

The standard library preparation protocol for Illumina mRNA sequencing used poly-dT beads to separate polyadenylated mRNA from other types of non-polyadenylated RNA such as rRNA and tRNA. For choanoflagellates, the bead selection step also served to separate choanoflagellate mRNA from bacterial RNA (which is not polyadenylated). Because the amount of bacterial RNA isolated from a culture often exceeded the amount of choanoflagellate RNA by one to several orders of magnitude, we reasoned that the standard bead selection might not be sufficient. We tested this hypothesis on *S. rosetta* Px1, a culture grown with a single species of bacterium, *Algoriphagus machipongonensis*. Because both species have sequenced genomes (*Alegado et al., 2011*; *Fairclough et al., 2013*), we could identify the origin of sequenced RNA using a straightforward read mapping procedure. We cultivated *S. rosetta* Px1 cells as described previously (*Fairclough et al., 2010*). We scraped and harvested 50 ml of culture after three days of growth in a 150 ml tissue culture dish. We performed centrifugation (with a 10 min spin), RNA isolation, DNase digestion and total RNA quality assessment as described above.

We compared the standard Illumina TruSeq v2 mRNA preparation protocol, which performs two rounds of polyA selection with a single set of poly-dT-coated beads, against a modified protocol that repeats the polyA selection steps, for a total of four rounds of polyA selection with two sets of beads. For all other aspects of library preparation, we followed the manufacturer's protocol. We quantified libraries by qPCR (Kapa Biosystems, Sigma-Aldrich) and sequenced them on a HiSeq 2000 machine (Illumina, San Diego, California, United States) at the Vincent J. Coates Genomics Sequencing Laboratory at the California Institute for Quantitative Biosciences (Berkeley, California, United States).

We generated 16,970,914 single-end 50 bp reads for the library prepared with two rounds of polyA selection, and 17,182,953 for the four-round library. We mapped reads to the *S. rosetta* and *A. machipongonensis* genomes using BWA version 0.6.1 (*Li and Durbin, 2009*) and SAMtools version 0.1.18 (*Li et al., 2009*) with default parameter values. We counted reads mapping to *S. rosetta* ribosomal loci on supercontig 1.8 (5S: positions 1900295–1900408, 18S: 1914756–1916850 and 28S: 1917502–1923701). The number of reads mapping to the non-ribosomal portion of the *S. rosetta* genome did not differ substantially between the two data sets: 12,737,031 reads mapped for the two-round data, and 12,585,647 for the four-round data. Similarly, 10,509,262 reads from the two-round data mapped to *S. rosetta* transcripts and 10,181,522 for the four-round data. We also asked whether additional rounds of polyA selection would cause increased RNA breakage due to pipetting or heating during the selection process [e.g., (*Kingston, 2001*)], leading to lower coverage of the 5' ends of transcripts (because the poly-dT sequence binds to the 3' end of RNA molecules). We estimated the loss of 5' transcript ends due to shear to affect less than roughly 5% of transcripts (*Figure 1—figure supplement 2a*).

The four-round method removed roughly an order of magnitude more non-polyadenylated RNA than the two-round method (*Figure 1—figure supplement 2b*). We observed that the four-round data set had a slightly lower overall read quality. To address this, we tested whether a difference in read quality between the two data sets could account for the difference in read mapping by randomly resampling the two-round data set to contain the same number of either Phred-like quality 20 or Phred-like quality 30 bases as the four-round data set, but neither resampling affected the results. We also tested whether transcript assembly quality would suffer in the four-round data set by assembling both data sets de novo with Trinity release 2012-03-17 (*Grabherr et al., 2011*) with default parameter values and mapping the resulting contigs to the *S. rosetta* genome using BLAT version 34 (*Kent, 2002*) with default parameter values, but we observed no substantial difference between the two data sets.

Given the superior ability of four rounds of polyA selection to remove contaminating bacterial RNA with little to no loss of transcript coverage, we adopted this methodology for subsequent transcriptome sequencing. The raw sequence reads for this experiment are available at the NCBI Short Read Archive with the BioProject identifier PRJNA420352.

## Library preparation and mRNA sequencing

We began the Illumina TruSeq v2 mRNA library preparation protocol with approximately 2 μg of total RNA per culture, if available (*Supplementary file 1*). We performed four rounds of polyA selection (instead of the standard two) and introduced two further modifications to the standard protocol: first, we repeated the Agencourt AMPure XP (Beckman Coulter, Indianapolis, Indiana, United States) bead clean-up step to enhance adapter removal, and second, we used 1.5 μl less volume in all bead elution steps, in order to reduce loss during the protocol. We prepared libraries from 5 RNA samples at a time, and the libraries were later multiplexed into groups of 6 or seven per sequencing lane. To allow us to detect evidence of potential cross-contamination during either process, we ensured that the groupings for sample preparation and sequencing were distinct (*Supplementary file 1*).

We estimated library concentration using the Qubit dsDNA HS Assay (Thermo Fisher Scientific) and determined quality and fragment size distribution with a Bioanalyzer 2100 High Sensitivity DNA chip (Agilent). We quantified libraries by qPCR (Kapa Biosystems, Sigma-Aldrich) and sequenced them on an Illumina HiSeq 2000 at the Vincent J. Coates Genomics Sequencing Laboratory at the California Institute for Quantitative Biosciences (Berkeley, California, United States). One group of libraries was sequenced twice (consisting of *A. spectabilis*, *Diaphanoeca grandis*, *Helgoeca nana*, *S. helianthica*, *S. infusionum* and *Salpingoeca urceolata*) due to a drop-off in quality after base 50 on the forward read of sequencing pairs; quality scores up to base 50 on the forward read and on reverse reads were not affected. The second, repeat sequencing run did not experience this issue. We incorporated both sequencing runs for affected libraries into subsequent analyses (including Quality trimming and Error correction, see below). We produced between 23 million and 61 million paired-end 100 bp sequencing reads per library (*Supplementary file 1*). Raw sequence reads are available at the NCBI Short Read Archive with the BioProject identifier PRJNA419411 (accession numbers for each species are listed in *Supplementary file 1*).

## Quality trimming

We trimmed sequence reads using Trimmomatic version 0.30 (*Lohse et al., 2012*) with two separate filters: (1) removal of TruSeq adapter sequence and (2) trimming very low quality bases from the ends of each read. To implement these filters, we ran Trimmomatic in three phases. In the first phase, we clipped palindromic adapters using the directive ILLUMINACLIP:2:40:15 and discarded resulting reads shorter than 25 bases with MINLEN:25. This resulted in two data sets: one containing reads whose mate pair remained in the set, and the other composed of reads whose pair was removed due to adapter contamination. In the second phase, we clipped simple adapters from the remaining paired data set using the directive ILLUMINACLIP:2:40:15, imposed a minimum Phred-like quality cutoff of 5 on the first ten and last ten bases using LEADING:5 and TRAILING:5, subjected the read to a minimum sliding window quality using SLIDINGWINDOW:8:5 and discarded resulting reads shorter than 25 bases with MINLEN:25. The third phase operated on the remaining unpaired reads from the first phase, and implemented the same directives as the second phase. We used a permissive minimum quality of 5 in order to remove very low quality bases, as these might interfere with read error correction in the subsequent processing step. We discarded reads less than 25 in length because they were shorter than the k-mer size of the Trinity assembler (see De novo transcriptome assembly below). In all adapter clipping operations, we used sequences appropriate to the index used for multiplexed sequencing. The number of sequence reads and total bases remaining after trimming for each library are given in *Supplementary file 1*.

## Error correction

We performed error correction on trimmed reads using Reptile v1.1 (*Yang et al., 2010*) following the authors' instructions, with the modifications described below. We began by using the 'fastq-converter.pl' script to convert from FASTQ and to discard reads with more than one ambiguous character 'N' in any window of 13 bases. For reads with one 'N', we chose the character 'a' as the substitute for 'N', as all of the characters in our input reads were in upper case (A, C, G, or T); thus, we could later recognize 'N' bases converted in this step. Next, we tuned parameters using the 'seq-analy' utility following the authors' instructions, in four steps: (1) Running 'seq-analy' with default settings. (2) Adjusting the input settings to 'seq-analy' using the results from step 1. For all

species, we set MaxBadQPerKmer to eight and KmerLen to 25 (to match the k-mer length used in Trinity). (3) Re-running 'seq-analy' using the adjusted input settings. (4) Creating the input settings to 'Reptile' based on the output of step 3. We set KmerLen to 13 and step to 12 for all species. The values of QThreshold, T_expGoodCnt, T_card and Qlb differed by species (*Supplementary file 1*). All other parameters were left at their defaults to run Reptile.

We noticed that the locations within reads of errors identified by Reptile fell into two general classes: sporadic errors not located adjacent to any other error, and clustered errors, in which several adjacent bases within the same k-mer window were identified as errors. In some extreme cases, every single base within a sequence read was identified as a target for error correction; we observed this phenomenon in the set of read corrections for every species. We reasoned that this was an unintended consequence of the iteration-to-exhaustion approach (step 2d) of the Reptile algorithm. Therefore, we designed a method to correct sporadic errors, but not clustered errors. For each species, we began by grouping each read according to the total number of errors identified. Within each group, we built a distribution of the number of adjacent errors within the same k-mer window. For sporadic errors, this number should be close to 0, but for clustered errors, the number could be up to the k-mer size minus one. There was a clear pattern within each of these distributions, with some errors identified with no neighbors (sporadic errors), a smaller number identified with one neighbor, and an increasing number beginning at two or more neighbors (clustered errors). We used these empirical distributions to set the maximum allowable amount of neighboring errors within a k-mer window as the count just prior to the beginning of the secondary increase within each distribution. For example, for *D. grandis*, in the case of the group of reads containing four total identified errors, there were 316,103 errors with no neighbors within the same k-mer, 197,411 with one neighbor, 156,043 with two neighbors, and 353,639 with three neighbors (that is, all four errors were within the same k-mer window). Thus, for the group of reads containing four total identified errors in *D. grandis*, we only corrected errors with up to two neighboring errors in the same k-mer window. After running Reptile error correction of sequence reads and quality files subject to these cutoffs, we performed a final step of restoring ambiguous bases converted by 'fastq-converter.pl' (from 'N' to 'a') that were not subsequently corrected by Reptile back to their original value of 'N'.

## De novo *transcriptome assembly*

We performed de novo transcriptome assembly on trimmed, corrected sequence reads and quality files with Trinity release 2013-02-25 (*Grabherr et al., 2011*) with '–min_contig_length' set to 150 and all other parameters at their default values. We chose a minimum contig length of 150 (rather than the default of 200) so as not to exclude assembly fragments that might encode predicted proteins with lengths between 50 and 66 amino acids, because some domains in the Pfam database are as short as 50 amino acids. Because none of the species we sequenced had an available genome assembly, we did not know whether transcripts might be encoded in overlapping positions within the genome. To test this possibility, we repeated each Trinity assembly with the addition of the '–jaccard-clip' option and compared the estimated number of fusion transcripts predicted by Transdecoder release 2012-08-15 (*Haas et al., 2013*). We found essentially no difference in the number of predicted fusion transcripts between the original and '–jaccard-clip' assemblies, and so we continued with the original assemblies. Assembly statistics are reported in *Supplementary file 1*.

## Identification and removal of cross-contamination

Cross-contamination within a multiplexed Illumina sequencing lane is estimated to cause incorrect assignment of roughly 0.5% of index pairs (*Kircher et al., 2012*). We designed a procedure to eliminate transcriptome assembly contigs resulting from incorrect index assignments. We ran blastn version 2.2.26 (*Altschul et al., 1997*) with a maximum E value of $1 \times 10^{-10}$ to query contigs from each species against all other species. Because of the evolutionary distances among the choanoflagellates we sequenced (*Figure 2—figure supplement 1*), most contigs from one species had no matches in any other species. Within the contigs that did have cross-species matches (*Figure 1—figure supplement 3a*), we observed a large number that were identical or nearly-identical, which were likely cross-contaminants, and another set of matches distributed around roughly 80% identity, likely representing highly conserved genes. The two cases were separated at roughly 96% identity. After exploring the distribution of match lengths in a similar manner (*Figure 1—figure supplement 3b*),

we considered matches at 96% or greater identity of at least 90 bases in length to be cross-contaminated.

Next, we identified the sources of cross-contaminated contigs by comparing the number of reads mapping from both species for each match. We first masked contigs with Tandem Repeats Finder version 4.04 (*Benson, 1999*), with the following parameter values: match = 2, mismatch = 7, indel penalty = 7, match probability = 80, mismatch probability = 10, min score = 30, max period = 24. We next mapped reads to masked contigs using the Burroughs-Wheeler Aligner, BWA, version 0.7.5a (*Li and Durbin, 2009*) and SAMtools version 0.1.18 (*Li et al., 2009*). We ran BWA 'aln' with the '-n 200' option to allow up to 200 equally best hits to be reported, and all other parameter values left at their defaults. Based on the distribution of read mapping ratios between the pair of species matching for each cross-contaminated contig (*Figure 1—figure supplement 3c*), we retained only contigs for the species in a pair with 10 times or more reads mapping, and discarded all other contigs, with one exception: if a contig had at least 10,000 reads mapping, we did not discard it, regardless of read mapping ratio. We observed that many contigs encoding conserved genes (for example, α-tubulin and elongation factor 1α) also tended to be the most highly expressed, and thus the read mapping ratio was often close to one for these contigs. We identified as cross-contaminated and removed between 1.7 and 8.8% of contigs for each species (*Supplementary file 1*). We note that our procedure would also be expected to discard sequences from any bacterial species that were present in two different choanoflagellate cultures. For a more detailed examination of the cross-contamination removal process, see (*Richter, 2013*).

## Prediction of amino acid sequences from assembled transcripts and elimination of redundant transcripts

We predicted proteins from decontaminated contigs with Transdecoder release 2012-08-15 (*Haas et al., 2013*) with a minimum protein sequence length of 50. We noticed that many of the proteins originating from different contigs within a species were completely identical along their entire length. Furthermore, we also observed many contigs whose predicted proteins were a strict subset of the predicted proteins from another contig. For example, contig one might encode predicted proteins A and B, and contig two might encode two predicted proteins exactly matching A and B, and a third predicted protein C. We removed both types of redundancy (exact matches and subsets) from the set of predicted proteins, and we also removed the contigs from which they were predicted (*Supplementary file 1*).

## Measurement of expression levels and elimination of noise transcripts

To estimate expression levels, we mapped sequence reads to decontaminated, non-redundant, Tandem Repeats-masked contigs using the Burroughs-Wheeler Aligner, BWA, version 0.7.5a (*Li and Durbin, 2009*). We ran BWA 'mem' with the '-a' option to report all equally best hits, with all other parameter values left at their defaults. We converted BWA output to BAM format using SAMtools version 0.1.18 (*Li et al., 2009*) and ran eXpress version 1.4.0 (*Roberts and Pachter, 2013*) with default parameter values in order to produce estimated expression levels, in fragments per kilobase per million reads (FPKM). The distribution of FPKM values (*Figure 1—figure supplement 3d*) had a peak near 1, with steep decreases in the number of contigs at lower values. Therefore, we chose an extremely conservative noise threshold two orders of magnitude below the peak, at FPKM 0.01, and discarded contigs (and their associated predicted proteins) below this value (*Supplementary file 1*).

The final sets of contigs are available as Dataset 1, and the proteins as Dataset 2 (*Richter et al., 2018*). FPKM values for contigs are given in Dataset 3 (*Richter et al., 2018*). We also submitted the final sets of contigs and proteins to the NCBI Transcriptome Shotgun Assembly (TSA) sequence database. The contigs (and their associated proteins) available in the TSA differ from our final sets in three ways: (1) The TSA database does not accept contigs with lengths less than 200, whereas our minimum was 150; (2) The submission system identified and rejected a small number of contigs as bacterial contaminants; (3) The submission system identified and required us to trim a small number of contigs to remove Illumina adapter sequences that were missed by our screen with Trimmomatic. Differences between the assemblies we analyzed and those submitted to NCBI TSA are summarized in *Supplementary file 1*, and a complete list of affected contigs can be found in Dataset 9 (*Richter et al., 2018*).

## Completeness of predicted protein sets

To determine whether the conserved gene content of the transcriptomes we produced was similar to the two sequenced choanoflagellate genomes, we searched our data for conserved eukaryotic proteins with BUSCO version 3.0.2 (*Simão et al., 2015*). We used default parameter values and the 303 BUSCOs from the 'eukaryota_odb9' set, and performed searches with HMMER version 3.1b2 (*Eddy, 2011*).

We note that each final transcriptome assembly contained between 18,816 and 61,053 proteins per species (*Supplementary file 1*), markedly more than the 9196 and 11,629 genes predicted, respectively, from the assembled genomes of *M. brevicollis* (*King et al., 2008*) and *S. rosetta* (*Fairclough et al., 2013*). The relatively higher protein counts predicted from choanoflagellate transcriptomes likely represent an overestimate resulting from the inherent complexities in assembling unique contig sequences from short read mRNA sequencing data in the absence of a reference genome (*Grabherr et al., 2011*), including the fact that sequence reads from different splice variants may have assembled into separate contigs while being encoded by the same gene. Because our goal was to reconstruct the full diversity of genes in the Urchoanozoan and Urmetazoan, the tendency of transcriptomes to yield overestimates of gene numbers was not a significant concern.

## Tests of choanoflagellate species identity and contamination with animal sequences

To confirm the identity of the choanoflagellate cultures we sequenced, we compared our transcriptome data to seven protein coding genes with choanoflagellate sequences previously available in GenBank: actin, alpha tubulin, beta tubulin, EF-1A, EFL, HSP70 and HSP90. To search the transcriptomes for each gene, we downloaded all previously available choanoflagellate sequences from GenBank, aligned them using FSA version 1.15.7 (*Bradley et al., 2009*) with default parameter values, and trimmed the alignments using Gblocks (*Talavera and Castresana, 2007*) with allowed gap positions set to 'half' and all other parameter values set to their most permissive. We next built HMMs for each trimmed alignment using hmmbuild from the HMMer package version 3.0 (*Eddy, 2011*) and searched the contigs from each transcriptome (and their reverse complements) with hmmsearch, both with default parameter values. In each case, the top hit we retrieved for each transcriptome (the contig with the lowest E value) matched the corresponding sequence for the species in GenBank.

We tested for the possibility of animal contamination within choanoflagellate transcriptomes using the same set of seven HMMs to search the nucleotide coding sequences from all 59 species in our data set. For each target species, we retained the top three hits by E value. For each gene, we aligned the resulting sequences using using MAFFT version 7.130b (*Katoh and Standley, 2013*) with the parameters '–maxiterate 1000 –localpair' and trimmed alignments using trimAl version 1.2rev59 (*Capella-Gutiérrez et al., 2009*) with the parameter '-gt 0.8'. We built phylogenetic trees with RAxML version 8.2.0 (*Stamatakis, 2014*) with the options '-m GTRCAT' to define the model of DNA substitution and '-f a -N 100' option for a rapid bootstrap analysis with 100 bootstraps. We collapsed branches with bootstrap support below 50 using Archaeopteryx version 0.9813 (*Han and Zmasek, 2009*). We observed no instances with ≥50% bootstrap support in which a choanoflagellate sequence was nested within a clade of animal sequences (*Figure 2—source data 1*), indicating that none of the choanoflagellate transcriptome sequences we retrieved appeared to be of animal origin.

## Construction of gene families and their probabilities of presence

To identify gene families, we chose a set of representative animals and outgroup species (*Supplementary file 3*). We used the 19 choanoflagellates we sequenced and the complete predicted protein sets from the *M. brevicollis* (*King et al., 2008*) and *S. rosetta* genomes (*Fairclough et al., 2013*). As an internal control, we had sequenced two independent isolates of *Stephanoeca diplocostata*, whose predicted proteins we compared using CD-HIT version 4.5.4 (*Li and Godzik, 2006*) with default parameter values. We found that the two *S. diplocostata* isolates contained essentially equivalent predicted protein sets, so we used only the French isolate for constructing orthologous groups. We compared the 21 choanoflagellate species to 21 representative animals with genome-scale sequence data available, with an emphasis on early-branching non-bilaterians: sponges, ctenophores, *T. adhaerens* and cnidarians. We included 17 outgroups with sequenced

genomes in our analysis: *C. owczarzaki*, a holozoan representative of the closest relatives of animals and choanoflagellates, five fungi chosen to represent fungal diversity, two amoebozoa, and nine species representing all other major eukaryotic lineages. The predicted proteins of the chytrid *Homolaphlyctis polyrhiza* were released in two partially redundant sets, which we combined using CD-HIT version 4.5.4 with default parameter values.

We applied OrthoMCL version 2.0.3 (*Li et al., 2003*) to construct gene families (orthologous groups of proteins) using recommended parameter values. As recommended in the OrthoMCL documentation, we performed an all versus all sequence homology search using blastp version 2.2.26 (*Altschul et al., 1997*) with an expectation value of $1 \times 10^{-5}$, and we determined orthologous groups with MCL-edge version 12–068 (*Enright et al., 2002*; *van Dongen and Abreu-Goodger, 2012*) using an inflation parameter of 1.5.

Our goal in constructing gene families was to identify groups composed of orthologous proteins. Although numerous approaches are currently available, no existing algorithm can yet identify all orthologous genes while perfectly separating orthologs from spurious BLAST hits (*Quest for Orthologs consortium et al., 2016*). We chose OrthoMCL due to its widespread use in previous studies, including analyses of animal gene family evolution based on the genomes of *S. rosetta* and *M. brevicollis* (*Fairclough et al., 2013*), and due to its relative balance of sensitivity and specificity in comparison to other algorithms (*Altenhoff and Dessimoz, 2009*).

Spurious BLAST hits within gene families pose a particular problem for ancestral gene content reconstruction. Approaches based on binary presence/absence calls, originally developed for morphological characters where the probability of homoplasy is considered to be much lower (*Farris, 1977*), may be vulnerable to yielding hyperinflated estimates of the gene content of the Urchoanozoan, since any gene family containing at least one protein from animals and one from choanoflagellates could be inferred to have been present in the Urchoanozoan. Indeed, we observed a number of gene families containing many animals but only one or two choanoflagellates, and vice versa (for example, there were 130 gene families containing one choanoflagellate and $\geq$15 animal species; *Figure 2—figure supplement 2a*). If these gene families represented true orthologous groups and were therefore present in the ancestral Choanozoan, they would require several independent loss events within one of the two groups; we reasoned that it was more parsimonious that some proportion of these genes evolved in only one of the two groups and that the isolated BLAST hits from the other group represented false positives.

To address this problem, we produced membership probabilities for each protein within each gene family, based on its average BLAST E value to all other proteins within the gene family (the absence of a BLAST hit between a pair of proteins was treated as the maximum possible (i.e., least probable) E value). A true ortholog should have low E value hits to nearly all other members of its gene family, resulting in a low average E value. In contrast, a false ortholog should have higher E value hits to only one or a few members of the gene family, resulting in a high average E value. If there were multiple proteins from the same species in a gene family, we chose the protein with the lowest average BLAST E value. To produce probabilities from average E values, we rescaled them using the empirical cumulative density function (*van der Vaart, 1998*, p. 265) of all E values from the initial all versus all homology search. We ordered E values from highest to lowest to produce a probability between 0 and 1 (*Figure 2—figure supplement 2b*). Within the resulting probabilities, a protein with low E value BLAST hits to all other members of its gene family had a probability close to 1, and a protein with high E value BLAST hits to only a few members of its gene family had a probability close to 0 (*Figure 2—figure supplement 2c–d*). We found that most proteins fell close to one of these extremes, and that the probabilities clearly distinguished hits between proteins within the same gene family versus hits between proteins in two different gene families (*Figure 2—figure supplement 2e*). In addition, proteins within gene families identified above containing many choanoflagellates and many animals had probabilities closer to 1, whereas some proteins within gene families with few choanoflagellates and many animals were assigned low probabilities, which are likely to be false orthologs, and some were assigned high probabilities likely representing true orthologs (*Figure 2—figure supplement 2f*). We conclude that this approach was able to identify and remove a substantial proportion of false orthologs, resulting in a set of gene families highly enriched for (although still not entirely composed of) truly orthologous families.

We tested the effectiveness of our approach to screen out false positive orthologs by building a phylogenetic tree using gene family 9066 as an example. This gene family contains numerous animal

sequences and a single choanoflagellate sequence, which was assigned a probability of 0.03 for gene family membership due to its high average blastp E value to other members of the family (*Figure 2—figure supplement 2c*). We first aligned the protein sequences in gene family 9066 using MAFFT version 7.130b (*Katoh and Standley, 2013*) with the parameters '–maxiterate 1000 –local-pair' and trimmed alignments using trimAl version 1.2rev59 (*Capella-Gutiérrez et al., 2009*) with the parameter '-gt 0.5'. We next built an HMM for the trimmed alignment using hmmbuild from the HMMer package version 3.0 (*Eddy, 2011*) and searched the full protein complement of each species using hmmsearch, both with default parameter values. For each species, we retrieved the top hit among sequences not present in gene family 9066. We aligned the sequences found by hmmsearch together with those from gene family 9066 using MAFFT and trimAl as above, modifying the trimAl missing sequences threshold to be '-gt 0.25', in order to include parts of the alignment only present in sequences from gene family 9066 (which represented slightly more than one quarter of the sequences included in the alignment). We built phylogenetic trees with RAxML version 8.2.0 (*Stamatakis, 2014*) with the options '-m PROTGAMMALGF' and '-f a -N 100'. We visualized the resulting tree and collapsed branches below 50% bootstrap support using the Interactive Tree of Life (iTOL) web site (*Letunic and Bork, 2016*). The choanoflagellate protein in gene family 9066 is more closely related to proteins outside the gene family than to proteins inside the family (*Figure 2—figure supplement 10*), and thus was correctly identified as a false positive by our probability assignment method.

Protein sequences for gene families are available as Dataset 4, and presence probabilities by species are listed as part of Dataset 5 (*Richter et al., 2018*).

## Inference of gene family origins and heat map

For each gene family, we calculated the sum of membership probabilities for species from each of the five major groups in this study (choanoflagellates, animals, *C. owczarzaki*, fungi, and other eukaryotes). Based on the resulting distribution (*Figure 2—figure supplement 3a*), for a gene family to be considered present in a major group, we required this sum to be ≥10% of the number of species in the major group (stated alternatively, we required the average presence probability in the group to be ≥10%). Notably, the 10% threshold was independent of the tree topology within each major group, thereby minimizing the impact of currently unresolved within-group species relationships [e.g., among early-branching animals (*King and Rokas, 2017*)]. For choanoflagellates and animals, each of which have 21 species in our data set, this equates, for example, to a gene family represented at high probability (≥70%, corresponding to an average BLAST E value of $1 \times 10^{-20}$; *Figure 2—figure supplement 2a*) in three or more species, or at 35% probability (E value of $1 \times 10^{-10}$) in six or more species. Therefore, the only truly orthologous gene families likely to be excluded by this threshold are those with weak homology (i.e., average BLAST E values $> 10^{-10}$) to only a few species in a major group, a rare case which would also require numerous independent losses of the gene family within the group.

Next, we developed a set of parsimony-based rules (*Supplementary file 4*) to determine the origin of each gene family. If a gene family was present within two separate major groups, we considered it to have been present in their last common ancestor. For gene families containing proteins from species in only one major group, we considered it to have been present in the last common ancestor of that major group only if it was present (i.e., satisfying the 10% probability threshold) within two or more separate sub-groups (*Figure 2—figure supplement 3b*). Within animals, we defined three sub-groups: sponges (*O. pearsei*, *Ephydatia muelleri*, *A. queenslandica*), ctenophores (*M. leidyi*), and later-branching animals (*T. adhaerens*, cnidarians, bilaterians). An animal-specific gene family present in at least two of these three sub-groups would be considered to have been present in the Urmetazoan (and thus to have evolved on the animal stem lineage); this inference would not change if either sponges or ctenophores were the earliest-branching animals. Similarly, a choanoflagellate-specific gene family was required to be present within both craspedids and loricates [two well-defined groups of species *Leadbeater, 2015*; *Carr et al., 2017*)] to be considered present in the Urchoanozoan. Inferred group presences for each gene family are available in Dataset 5 (*Richter et al., 2018*).

In the text, we precede estimated gene family counts with a tilde. We selected a conservative probability threshold of 10% in order to minimize the number of gene families that might erroneously be assigned as specific to a given group (e.g., animals or choanozoans). This choice may have

resulted in some gene families that are truly specific to a group instead being incorrectly assigned as shared with another group. As a consequence, counts of gene families originating in different groups should represent relatively conservative estimates.

To produce a heat map for visual display of gene families, we ordered them by their patterns of presence within the five major species groups. Within a given pattern (for example, absent in outgroups and fungi but present in *C. owczarzaki*, animals and choanoflagellates) we clustered gene families by Pearson correlation using Cluster 3.0 (*de Hoon et al., 2004*), with all other parameter values set to their defaults. We visualized heat maps using Java TreeView version 1.1.6r4 (*Saldanha, 2004*) and color palettes from ColorBrewer (*Harrower and Brewer, 2003*). For display purposes, we restricted *Figure 2* to gene families inferred to have been present in at least one of six nodes representing last common ancestors of interest: Ureukaryote, Uropisthokont, Urholozoan, Urchoanozoan, Urchoanoflagellate, and Urmetazoan. The full heat map for all gene families with representatives from at least two species is shown in *Figure 2—figure supplement 4*. Gene families present, gained and loss at each ancestral node are listed in Dataset 6 (*Richter et al., 2018*).

## Test of existing methods of ancestral gene content reconstruction

We tested three classes of existing ancestral reconstruction methods using presence/absence data as input: Dollo parsimony, maximum likelihood, and Bayesian. The three different methods produced substantially different estimates of ancestral gene family content (*Supplementary file 13*). We performed Dollo parsimony analysis on presence/absence data for all gene families using PHYLIP version 3.695 (*Felsenstein, 2013*) with default parameter values. For comparison to previous studies, we also gathered Dollo parsimony-based gene content estimates from *Fairclough et al., 2013*, which included two choanoflagellate species (*M. brevicollis* and *S. rosetta*). We ran maximum likelihood analysis on presence/absence data for all gene families using Mesquite version 3.40 (*Maddison and Maddison, 2018*). When running Mesquite, we supplied the phylogenetic tree shown in *Figures 2* and *3* as input (see Materials and methods, 'Species tree and phylogenetic diversity') and analyzed gene family content evolution with the AsymmMk (asymmetrical Markov k-state two parameter) model. We performed Bayesian analysis with MrBayes version 3.2.6 (*Ronquist and Huelsenbeck, 2003*). We supplied presence/absence data only for gene families present in more than one species (i.e., we eliminated singletons), and specified the options 'noabsencesites' and 'nosingletonpresence' to correct for unobservable site patterns following (*Pisani et al., 2015*), with *Naegleria gruberi* as the outgroup, gamma-distributed rate variation among sites, and other parameters left at their defaults. We ran separate analyses for each ancestral node of interest, each time including the species descended from that node as a topology constraint. We ran each analysis with a sampling frequency of 1000 generations until the average standard deviation of split frequencies between runs reached 0.01 (between 5 million and 10 million generations, depending on the ancestral node of interest). For the two analyses that produced probabilities of gene family presence at internal nodes (maximum likelihood and Bayesian), we calculated total counts as follows: for gene family presence, we summed the probabilities across all gene families. For gene family gains, we summed the difference in probability between each node of interest and its parent (e.g., the Urmetazoan and its parent, the Urchoanozoan) only for the gene families with a higher probability of presence in the node of interest compared with its parent. For gene family losses, we performed an analogous sum only for the gene families with lower probabilities in the node of interest in comparison to its parent.

## Test of gene family presence in recently released genomes

Because new genome sequences are continuously becoming available, we tested whether the ancestral gene content reconstruction we performed would be influenced by the addition of new genome-scale data from two key sets of species: early-branching animals and early-branching holozoans. We chose species with high-quality publicly available protein catalogs that would maximize the phylogenetic diversity added to our data set. For early-branching animals, we selected two sponges, the demosponge *Tethya wilhelma* version v01_augustus_prots (*Francis et al., 2017*) and the calcareous sponge *Sycon ciliatum* version SCIL_T-PEP_130802 (*Fortunato et al., 2014*), and one ctenophore, *Pleurobrachia bachei* version 02_P-bachei_Filtered_Gene_Models (*Moroz et al., 2014*). For early-branching holozoans, we selected the teretosporeans *C. fragrantissima*, *Ichthyophonus*

*hoferi*, *Chromosphaera perkinsii* and *Corallochytrium limacisporum* (*Grau-Bové et al., 2017*). Because we planned to implement a best reciprocal BLAST approach, which might be confounded by paralogs present within any of the species, we began by removing redundancy separately for each species using CD-HIT version 4.6 (*Li and Godzik, 2006*) with default parameter values. We then performed best reciprocal blastp from the protein catalog of each additional species versus all 59 species included in our original analysis (*Supplementary file 3*), with the same maximum E value ($1 \times 10^{-5}$). We retained only top reciprocal blastp hits. We next calculated gene family presence probabilities for each additional species, using a slightly modified procedure. Because we retained only the top hit for each additional species protein to each original species, an additional species protein could only match to a single representative per species within each gene family, although the family might contain multiple representatives per species. Thus, instead of calculating average E value from each additional species protein to all members of each gene family (as we did for the original set), we calculated the average of best hits to each species within the family. Next, because each protein from an additional species might hit multiple different gene families, we chose the single gene family match with the lowest average E value. Finally, for each gene family and each additional species, we selected the lowest average E value of any protein in the species to the gene family and translated those to probabilities using the same empirical cumulative density function as for the original analysis (*Figure 2—figure supplement 2b*).

To visualize whether the additional species might substantially impact our inferences of ancestral gene content, we inserted their presence probabilities into the original heat map of *Figure 2* without reordering gene families (*Figure 2—figure supplement 5*). The additional sponges display similar patterns of gene family presence probability to the original sponge species in the analysis, as does the additional ctenophore in comparison to the original ctenophore. With these species added, the number of gene families present in the Urmetazoan in our original data set, ~8,418, would increase by 37, sixteen of which would be previously choanoflagellate-specific gene families newly classified as shared with animals. Within the early-branching holozoans, *C. perkinsii* appears to show slight evidence for the presence of animal-specific gene families, but generally with low probabilities, roughly comparable to the choanoflagellate *S. dolichothecata*. Thus, we estimate that a very small number of animal-specific gene families would instead be classified as originating in Holozoa with the addition of early-branching holozoans (13 of ~1,944). In addition, since we only included one representative early-branching holozoan (*C. owczarzaki*) in our original analysis, a subset of gene families originally classified as originating in Choanozoa would instead be assigned to Holozoa with the additional species (30); in general, however, we did not strongly emphasize the distinction between choanozoan-origin and holozoan-origin gene families in our results. Furthermore, as described below (Protein domains and evidence for gene presence based on domain architecture), the additional species would have a negligable effect on our inferences of gene family presence based on protein domain architecture.

## Species tree and phylogenetic diversity

We attempted to produce a robustly supported phylogenetic tree reflecting the relationships among *M. brevicollis*, *S. rosetta* and the choanoflagellate species we sequenced. As input, we selected different subsets of gene families based on different criteria (for example, one set of criteria selecting the 49 gene families missing at most five species, with remaining species having exactly one copy, or another set of criteria selecting the 24 gene families present in all species, with no species having more than two copies and at most 10 species having two copies). Within each subset, we separately aligned each gene family using MAFFT version 7.130b (*Katoh and Standley, 2013*) with the parameters '–maxiterate 1000 –localpair' and trimmed alignments using trimAl version 1.2rev59 (*Capella-Gutiérrez et al., 2009*) with the parameter '-gt 0.5'. We then concatenated the trimmed alignments from each gene family and built trees with maximum likelihood (RAxML version 8.2.0 (*Stamatakis, 2014*) with the options '-m PROTCATGTR' and '-f a -N 100') and Bayesian (PhyloBayes-MPI version 1.5a (*Lartillot et al., 2013*) with the option '-dc') methods. We found that two species resting on long terminal branches, *S. dolichothecata* and *C. hollandica*, had low support values and inconsistent phylogenetic positions depending on the gene family subset and the analysis method used. We hypothesized that this might result from the fact that both lack sister species in our transcriptome data set. To test this, we instead built phylogenetic trees with the addition of nucleotide sequences from sister species for *C. hollandica* (*Codosiga botrytis* and *Sphaeroeca volvox*) and *S.*

*dolichothecata* (*Salpingoeca tuba*). For this analysis, we were limited by the small number of genes that had previously been sequenced for these species (a maximum of 5). We found that the phylogenetic positions of both *C. hollandica* and *S. dolichothecata* stabilized when their sister species were added [and that the choanoflagellate topology was consistent with that of (*Carr et al., 2017*)]. We concluded that a robust phylogenomic tree of choanoflagellates, with stable, supported positions for *C. hollandica* and *S. dolichothecata* may require further genome-scale sequencing from sister groups of both species.

Therefore, since the focus of this study was not phylogenetic tree construction, and because the topic has been addressed elsewhere, we selected species trees from prior publications. Furthermore, because our major findings were based on comparisons among groups whose phylogenetic relationships are well established (animals, choanoflagellates, *C. owczarzaki*, fungi and other eukaryotes), differences in tree topology within any of these groups should be of minor importance. For choanoflagellates, we used the species tree from (*Carr et al., 2017*), for animals (*Philippe et al., 2009*) and for other eukaryotes (*Burki et al., 2016*). Because the branching order of early animals is still under active debate, we depicted the relationships among early-branching animals as a polytomy. We visualized the resulting tree with Archaeopteryx version 0.9813 (*Han and Zmasek, 2009*).

To measure phylogenetic diversity, we selected a set of 49 gene families for which each species had exactly one protein representative and no more than five species were missing (or roughly 10% of the 59 species in our data set). We aligned the protein sequences in each gene family separately using MAFFT version 7.130b (*Katoh and Standley, 2013*) with the parameters '–maxiterate 1000 –localpair'. We performed two rounds of alignment trimming with trimAl version 1.2rev59 (*Capella-Gutiérrez et al., 2009*): in the first round, we used the parameter '-automated1', and we supplied the output of the first round to the second round, with the parameter '-gt 0.5'. We constructed phylogenetic trees from the two-round trimmed alignments separately for each gene family with RAxML version 8.2.0 (*Stamatakis, 2014*), with the options '-m PROTGAMMALGF' and '-f a -N 100'. We measured cophenetic distances separating pairs of species on the resulting trees (in units of substitutions per site along branches) using the ape package version 4.1 (*Paradis et al., 2004*) and plotted pairwise distances averaged across all 49 gene families using the ggplot2 package version 2.2.1 (*Wickham, 2009*), both with R version 3.4.1 (*R Core Team, 2017*) in the RStudio development platform version 1.0.143 (*RStudio Team, 2016*).

## Group-specific core gene families found in all extant members

To identify animal-specific or choanoflagellate-specific gene families that were also present in all species within either group, we required every species in the group to pass the 10% minimum probability criterion. These core gene families are subject to several potential technical artifacts. First, an incomplete genome or transcriptome assembly could result in a species appearing to lack a gene family. Second, gene families containing repeated or repetitive protein domains (e.g. EGF or Ankyrin) might cause inappropriate inferences of sequence homology in our BLAST-based approach. Third, a true orthologous gene family present in all animals could be erroneously partitioned by OrthoMCL into two or more gene families, neither of which would then be considered present in all animals. Fourth, a gene family which duplicated on the lineage leading to the last common ancestor of a group could produce two gene families, among which paralogs are incorrectly partitioned, resulting in one or more species appearing to lack one of the two families. Thus, the lists of core gene families should not be considered exhaustive, especially for serially duplicated or repeat-containing gene families.

We annotated core animal-specific gene families (*Table 1*) by selecting consensus features and functions in UniProt (*The UniProt Consortium, 2017*), FlyBase (*Gramates et al., 2017*) and WormBase (*Lee et al., 2018*). Notably, BTB/POZ domain-containing protein 9, whose function is relatively poorly characterized in comparison to other core animal-specific gene families, contains the BTB Pfam domain, which was identified as part of an expanded repertoire of ubiquitin-related domain architectures in animals (*Grau-Bové et al., 2015*).

## Gene family retention

To determine retention of ancestral gene families within extant species, we began with the sets inferred to have evolved on the stem lineages leading to the Ureukaryote, Uropisthokont,

Urholozoan, Urchoanozoan, Urchoanoflagellate, and Urmetazoan (*Supplementary file 4*, Dataset 6; *Richter et al., 2018*). For each set of gene families partitioned by ancestral origin, we summed the membership probabilities for each species. Because we applied a 10% membership probability threshold to consider a gene family to be present within group of species (see above, Inference of gene family origins and heat map), a species in one group might have a small residual sum of membership probabilities for gene families assigned to another group. As an example, some choanoflagellates may display a small amount of retention of animal-specific gene families, which represents the sum of non-zero membership probabilities that did not reach the 10% threshold.

To test whether *B. floridae, N. vectensis*, and *O. pearsei* retained the same gene families, we applied the 10% presence probability threshold within each species. We found that there were 7,863 Urmetazoan gene families retained in at least one of the three species and 5,282 retained in all three (67%).

Some animal species with draft genomes, for example *P. pacificus*, were among those that retained the fewest ancestral gene families. However, the lack of gene predictions resulting from an incomplete genome assembly is likely not as strong as the signal produced by gene loss. In the example case of *P. pacificus*, its sister species *C. elegans*, which has a finished genome, retains the second fewest gene families. We produced phylogenetic tree-based visualizations using the Interactive Tree of Life (iTOL) web site (*Letunic and Bork, 2016*). We produced bar charts using the ggplot2 package version 2.2.1 (*Wickham, 2009*) with R version 3.4.1 (*R Core Team, 2017*) in the RStudio development platform version 1.0.143 (*RStudio Team, 2016*).

## Gene family annotation and pathway analysis

We annotated gene family function using the PANTHER Classification System (*Thomas et al., 2003*). We used the PANTHER HMM library version 7.2 (*Mi et al., 2010*) and the PANTHER HMM Scoring tool version 1.03 (*Thomas et al., 2006*) with default parameter values and the recommended expectation value cutoff of $10^{-23}$. We used PANTHER-provided files to associate PANTHER HMM hits with Gene Ontology (GO) terms (*Ashburner et al., 2000*). Annotations for both sets of terms, for all input proteins (including those not placed into gene families) are available in Dataset 7 (*Richter et al., 2018*). Annotations by gene family are listed in Dataset 8 (*Richter et al., 2018*).

We compared pathways present in the Urholozoan, Urchoanozoan, Urchoanoflagellate and Urmetazoan using MAPLE version 2.3.0 (*Takami et al., 2016*). As input, we provided protein sequences from gene families present at each ancestral node, but only for species descending from the ancestral node (for example, for the Urholozoan, we only included sequences from holozoans). We supplied the following parameters to MAPLE: NCBI BLAST; single-directional best hit (bi-directional best hit would not have been appropriate, since our input database consisted of gene families containing closely-related proteins from multiple species); KEGG version 20161101; and organism list 'ea' (all eukaryotes in KEGG). We compared completeness based on the WC (whole community) module completion ratio. We listed modules which differed by 25% or greater in completeness between the Urchoanozoan and Urmetazoan in *Supplementary file 8*, between the Urholozoan and Urchoanozoan in *Supplementary file 10*, and between the Urchoanozoan and Urchoanoflagellate in *Supplementary file 9*. To focus on amino acid biosynthesis pathways (*Figure 2—figure supplement 7*), we exported KEGG results from MAPLE for the Urchoanozoan and Urmetazoan gene sets and compared them using the KEGG Mapper Reconstruct Pathway tool, pathway ID 01230 (*Kanehisa et al., 2012*).

## Evidence for gene presence based on protein domain architecture

We predicted protein domains with the Pfam version 27.0 database and pfam_scan.pl revision 2010-06-08 (*Punta et al., 2012*), which used hmmscan from the HMMER 3.0 package (*Eddy, 2011*). We performed all Pfam searches with default parameter values. We predicted signal peptides and transmembrane domains using Phobius version 1.01 (*Käll et al., 2004*) with default parameter values. Domains for all input proteins (including those not placed into gene families) are listed in Dataset 7 (*Richter et al., 2018*). Domains by gene family are shown in Dataset 8 (*Richter et al., 2018*).

To determine which animal-specific gene families lacked a Pfam domain of known function, we calculated the proportion of proteins in each animal gene family that were annotated with a given Pfam domain (*Figure 2—figure supplement 11*). We accepted Pfam domains present in at least

10% of proteins in a gene family. Domains of unknown function in the Pfam database had names beginning with 'DUF'. To ensure that these domain names were not assigned a function in a more recent version of the Pfam database published after our initial annotation, we checked against Pfam version 31.0 and considered all domains whose names no longer began with 'DUF' as having an assigned function.

We established a set of criteria defining 'strong', 'moderate' and 'weak' evidence for the presence of genes of interest, based on domain content and OrthoMCL gene families; all criteria can be found in *Supplementary file 6*. In all cases, 'strong' evidence for conservation consisted of the canonical domain architecture (a diagnostic domain or set of domains in a specific order). Because Pfam domains are constructed from a set of aligned protein sequences selected from sequenced species, they are often strongly enriched for animals (and, in particular, for animal models such as *D. melanogaster* and *C. elegans*) and therefore may be biased against detecting instances of protein domains with more remote homology. To address this concern, for some gene families, we considered the presence of a protein in the same gene family as another protein with 'strong' evidence to constitute potential 'moderate' or 'weak' evidence. In addition, we considered previous reports to be 'strong' evidence in two cases: the presence of a canonical TLR in *N. vectensis* (*Miller et al., 2007*; *Sullivan et al., 2007*), and the presence of NF-κB in *C. owczarzaki* (*Sebé-Pedrós et al., 2011*).

Genome or transcriptome data for nine additional early-branching holozoans became available since our initial analyses of protein domain architecture (*Torruella et al., 2015*; *Grau-Bové et al., 2017*). To test whether the genes and protein domains of interest that we inferred to have originated in the Urchoanozoan did not in fact originate in the Urholozoan, we interrogated the domain content of the nine new species. In all cases, there were no proteins in any of the nine species with domain architectures that would qualify as 'strong' evidence to change our inference of ancestral origin (i.e., where there was not already 'strong' evidence in *C. owczarzaki*).

## Notch and Delta

To identify Notch and Delta, we relied solely on conserved protein domain architecture rather than on BLAST-based or phylogenetic evidence. In addition to their diagnostic protein domains, both families contain repeated, common protein domains. These can result in false inferences of homology for BLAST, and in difficulties in producing robust sequence alignments for phylogenetic analysis. Previous studies on Notch and Delta encountered difficulties in producing resolved phylogenetic trees, which were more pronounced for early-branching animals (*Gazave et al., 2009*) and potentially due to the presence of EGF repeats (*Rasmussen et al., 2007*). To test these effects in our data, we built phylogenetic trees for both Notch and Delta. For Notch, we selected all proteins encoding a Notch domain, and for Delta, all proteins encoding a DSL (Delta Serrate ligand) domain. We aligned Notch and Delta separately using MAFFT version 7.130b (*Katoh and Standley, 2013*) with the parameters '–maxiterate 1000 –localpair' for high accuracy alignments, trimmed them using trimAl version 1.2rev59 (*Capella-Gutiérrez et al., 2009*) with the option '-gt 0.4' to discard positions represented in fewer than 40% of proteins in the alignment, and built phylogenetic trees using RaxML version 8.2.0 (*Stamatakis, 2014*), with the options '-m PROTGAMMALGF' to define the model of protein substitution and '-f a –N 100' for a rapid analysis with 100 bootstraps. We visualized trees and removed branches with less than 50% bootstrap support using the Interactive Tree of Life (iTOL) web site (*Letunic and Bork, 2016*), yielding trees that were largely unresolved for both genes (*Figure 2—figure supplement 12*), consistent with previous results and reinforcing the use of protein domain architecture to identify Notch and Delta.

For Notch, we defined 'strong' evidence for conserved domain architecture to be one or more EGF repeats, a Notch domain, a transmembrane domain and an Ank domain, in that order. This domain architecture was unique to animals and choanoflagellates in our data set. Many, but not all, Notch genes in animals also contain NOD and NODP protein domains. However, we did not use these as evidence, because numerous animal Notch genes do not encode these domains, including some bilaterians (*Gazave et al., 2009*) (for example, the primate *Macaca mulatta* (UniProt ID F7HCR6), the tapeworm *Hymenolepis microstoma* (A0A068XGW6), and the nematode *Ascaris suum* (U1NAR2)). In our data, the NOD and NODP domains were not present in any sponge or choanoflagellate species, although two of the three sponges (*E. muelleri* and *A. queenslandica*) encoded proteins with EGF, Notch, transmembrane and Ank domains, and a previous study found that *A.*

*queenslandica* Notch and Delta were both expressed in in patterns consistent with other animals (*Richards et al., 2008*). We hypothesize that the NOD and NODP domains were not detected in either choanoflagellates or sponges because the NOD and NODP Pfam models were constructed from sequence alignments that included proteins mostly from bilaterians.

For Delta, we defined 'strong' evidence as the presence of both an MNNL (N terminus of Notch ligand) and a DSL domain. Both of these domains were considered to be animal-specific prior to this study [although we note that the DSL domain is found in two recently sequenced holozoans, the ich-thyosporeans *C. fragrantissima* and *Pirum gemmata* (*Grau-Bové et al., 2017*)]. The choanoflagellate *S. dolichothecata* expressed a protein encoding both the MNNL and DSL domains (m.249548), but lacking the transmembrane domain typically found in Delta. However, the contig encoding this protein was only partially assembled, and the presence of other partially assembled contigs in *S. doli-chothecata* encoding combinations of DSL, transmembrane and EGF domains increases our confidence that *S. dolichothecata* encodes a *bona fide* Delta. (Predicted proteins encoding MNNL and DSL, but not a transmembrane domain, were also present in some animals in our data set; for example, *Capitella teleta* and *Strongylocentrotus purpuratus*.)

## TGF-β signaling pathway

The canonical TGF-β signaling ligand (e.g., Activins or BMPs) is characterized by the TGFb_propep-tide (LAP) and TGF_beta domains (*Munger et al., 1997*), the type I TGF-β receptor by the combina-tion of a TGF_beta_GS and a kinase domain, and the transcriptional activator SMAD by the MH1 and MH2 domains (*Heldin et al., 1997*). Although none of these combinations was present in choa-noflagellates, we detected the individual TGFb_propeptide, TGF_beta_GS, MH1 and MH2 domains encoded in separate proteins (*Figure 2—figure supplement 9*). None of these four domains is found in eukaryotes outside Choanozoa (kinases are ancient domains, and the TGF_beta domain is found only in animals).

## Toll-like receptor signaling and innate immunity

For our analysis of Toll-like receptors, we considered all proteins containing two of the three follow-ing Pfam domains: LRR, Pkinase and TIR, as these are found in immune receptors and signaling pro-teins in both animals and plants. We found that the domain architecture of LRR, transmembrane domain and TIR, in that order, was unique to choanozoans, consistent with the independent evolu-tion in plants of immune-related proteins that contain the same protein domains but with distinct domain architectures, different pattern recognition, and different downstream signaling pathways (*Ausubel, 2005*; *Leulier and Lemaitre, 2008*). The domain architectures LRR, transmembrane, Pki-nase, TIR ('Kinase TLRs') and Pkinase, TIR lacking a transmembrane domain ('Kinase TIRs') were both unique to choanoflagellates. We did not distinguish among members of the LRR and TIR families (whose names in Pfam take the form of LRR_X, where X is a number, or TIR/TIR_2), because we found that different animal TLRs could contain domain combinations of each type. For example, *M. musculus* TLR11 (UniProt Q6R5P0, NCBI GI 45429992) contains LRR_8 and TIR_2 domains, whereas *M. musculus* TLR6 (UniProt Q9EPW9, NCBI GI 157057101) contains LRR_4 and TIR domains. We also did not attempt to differentiate among single vs. multiple cysteine cluster LRR domains (*Imler and Zheng, 2004*), because we found that TLRs containing a multiple cysteine cluster LRRNT Pfam domain as the final LRR prior to the transmembrane region were restricted to *D. melanogaster* within our data set.

We tested whether phylogenetic evidence could be used to confirm or contradict our findings based on protein domain architecture, although we anticipated that the repeated nature of LRR domains and the relatively short length of TIR domains might interfere with the resolution of phylog-eny reconstructions. We aligned all proteins in our data set containing two of the three of LRR, Pki-nase and TIR with MAFFT version 7.130b (*Katoh and Standley, 2013*) with default parameters (we were unable to build an alignment using MAFFT high accuracy parameters due to the large number of proteins involved). We trimmed the alignment using trimAl version 1.2rev59 (*Capella-Gutiérrez et al., 2009*) with the option '-gt 0.4' and built phylogenetic trees using RAxML version 8.2.0 (*Stamatakis, 2014*), with the options '-m PROTGAMMALGF' and '-f a -N 100'. We visualized trees and removed branches with less than 50% bootstrap support using the Interactive Tree of Life (iTOL) web site (*Letunic and Bork, 2016*). The resulting tree was largely unresolved, as anticipated

(*Figure 4—figure supplement 2*), although there were no cases in which a choanozoan-type TLR (LRR, transmembrane, TIR) was found in the same supported clade as a plant protein, consistent with the hypothesis that they evolved independently.

Although we did not find strong evidence for canonical animal TLR domain architecture in sponges, *M. leidyi* and *T. adhaerens*, we did detect proteins in sponges and cnidarians matching the domain content of animal interleukin receptors, which can also signal via MyD88 (*O'Neill and O'Neill, 2008*): an extracellular immunoglobulin (Ig) domain, a transmembrane domain, and an intracellular TIR domain [as found previously for the sponge *A. queenslandica* (*Gauthier et al., 2010*)]. This architecture was not present in choanoflagellates nor any other non-animal in our data set. We note that, in our analyses, the proteins previously identified as TLR2 homologs in *A. queenslandica*, *O. carmela* and other sponges (*Riesgo et al., 2014*) appear to have domain architectures matching interleukin receptors or other proteins, but not TLRs.

For genes in the canonical *M. musculus* TLR/NF-κB signaling pathway, we collected information on domain architectures in the Pfam 31.0 database (which did not include the choanoflagellates sequenced in this study) and on the species present in the same gene family as the *M. musculus* protein (*Supplementary file 14*). With the exception of adapters and kinases containing the Death domain, there were no architectures that were both specific to animals and present across animal diversity, two features required of diagnostic architectures for a gene family. Similarly, we did not identify any gene families restricted to holozoans (as would be expected for genes participating in the TLR/NF-κB pathway) and whose choanoflagellate members we could unequivocally identify as orthologous to the *M. musculus* proteins in the family. Thus, we restricted our current analyses to TLRs, adapters containing TIR or Death domains, and NF-κB. Future analyses more focused on detecting individual signaling components [e.g., (*Gilmore and Wolenski, 2012*)] are likely to further elucidate their evolutionary histories.

We built an alignment for all proteins in the gene family containing SARM1 (gene family 6840), with the addition of the human sequence (UniProt Q6SZW1), since previous work has focused on the human SARM1 protein. We aligned proteins with MAFFT 7.130b (*Katoh and Standley, 2013*) with the parameters '–maxiterate 1000 –localpair' for high accuracy alignments. We visualized the alignment with JalView 2 (*Waterhouse et al., 2009*). In addition to SARM1 and Kinase TIRs, choanoflagellates also encoded a diversity of protein domains associated with intracellular TIR domain-containing proteins (lacking a transmembrane domain). Domains found associated with TIR in multiple choanoflagellate species included ankyrin repeat (Ank) and armadillo (Arm) protein-protein interaction domains and Src homology 2 (SH2) kinase signal transduction domains.

In addition to TLRs, we also searched the choanoflagellates in our data set for cytosolic immune sensing genes. Several classes of nucleotide-binding domain and leucine-rich repeat (NLR) proteins have previously been detected in the *A. queenslandica* genome, suggesting that NLRs were present in the Urmetazoan (*Yuen et al., 2014*). The architecture of NLRs consists of a protein interaction domain (either Death or CARD), a nucleotide binding domain (NACHT) and multiple LRRs. Although we detected proteins encoding both NACHT and LRR domains in multiple choanoflagellates, neither the Death or CARD domain was present in any choanoflagellate we studied. The CARD domain in animal NLRs either directly or indirectly mediates activation of caspases (*von Moltke et al., 2013*), which are among the 39 core animal-specific gene families that we detected, suggesting that both NLRs and associated caspase downstream signaling activity may be animal innovations. Plants possess an analogous intracellular sensing pathway, NBS-LRR, which, like the TLR pathway, is thought to have evolved independently (*Urbach and Ausubel, 2017*). Diagnostic domains were absent in choanoflagellates for four additional cytosolic sensors: three containing CARD domains [mitochondrial antiviral signaling protein (MAVS), melanoma differentiation-associated protein 5 (MDA5) and retinoic acid inducible gene 1 (RIG-I)] and one containing HIN and PYRIN domains [absent in melanoma 2 (AIM2)]. However, we did find evidence for two other sensors previously reported in *M. brevicollis* (*Wu et al., 2014*): STING is present in both *Salpingoeca macrocollata* and *M. brevicollis*, based on the presence of the diagnostic Pfam domain TMEM173, and cGAS, identified by the Pfam domain Mab-21, is present in *S. macrocollata*, *M. brevicollis* and *M. fluctuans*.

## RNAi machinery in choanoflagellates

To define 'strong' evidence for the presence of Argonaute in our data set, we used the conserved Pfam domain architecture DUF1785, PAZ and Piwi (*Supplementary file 6*). To detect 'strong'

evidence for Dicer, we searched for the architecture Dicer_dimer, PAZ and Ribonuclease_3 (we considered the Dicer_dimer domain alone to be 'moderate' evidence). We did not detect Piwi in choanoflagellates, although it is present in most other eukaryotic lineages. Piwi is thought to repress transposable elements (*Aravin et al., 2001*; *Aravin et al., 2007*), and *M. brevicollis*, the only choanoflagellate species that has been investigated for transposable elements, appears to have very few (*Carr et al., 2008*).

Argonaute and Dicer were each lost five times independently in choanoflagellates (*Figure 3—figure supplement 2*), although these could both be overestimates if the genes were not expressed in our culture conditions. A similar pattern of repeated parallel loss has also been previously observed in kinetoplastids, a group of single-celled eukaryotes (*Matveyev et al., 2017*). Curiously, in contrast to the case for kinetoplastids, where Argonaute and Dicer were generally lost together, we detect five choanoflagellate species in which Argonaute is present and Dicer is absent or vice versa. These absences could reflect the presence of non-canonical RNAi genes or Dicer-independent RNAi pathways in choanoflagellates, as previously reported in *Saccharomyces cerevisiae* and in *M. musculus*, respectively (*Drinnenberg et al., 2009*; *Cheloufi et al., 2010*).

## Cell adhesion: Flamingo, Protocadherins, and Integrins

Diverse cadherins have previously been identified in choanoflagellates (*Abedin and King, 2008*; *Nichols et al., 2012*), but two classes of cadherins involved in animal planar cell polarity and development were thought to be animal-specific: Flamingo and Protocadherins.

The first described Flamingo cadherin (also known as starry night) is involved in the Frizzled-mediated establishment of planar polarity in the *D. melanogaster* wing (*Chae et al., 1999*; *Usui et al., 1999*). In animals, Flamingo cadherins are distinguished by the combination of three diagnostic Pfam domains, the presence of which constituted 'strong' evidence for Flamingo in our data set (*Supplementary file 6*): Cadherin, GPS and 7tm_2, a seven-pass transmembrane domain. The transcriptome of *C. hollandica* is predicted to encode a protein with all three diagnostic domains (*Figure 2—figure supplement 6*). In addition to these three domains, many animal Flamingo cadherins also contain HRM, EGF, and Laminin domains. Nine further choanoflagellate species express proteins with GPS, 7tm_2, and one of these additional domains ('moderate' evidence), and all choanoflagellate species express proteins with GPS and 7tm_2 domains ('weak' evidence).

Protocadherins are a large and diverse family of genes involved in animal development and cell adhesion (*Frank and Kemler, 2002*), and most, but not all, family members possess a diagnostic transmembrane Protocadherin domain paired with one or more extracellular Cadherin domains. Five species of choanoflagellates expressed proteins with a Protocadherin domain ('strong' evidence for conservation, *Figure 2—figure supplement 6*), although none of these also contained a Cadherin domain. The *Choanoeca perplexa* Protocadherin domain-containing protein also contains a transmembrane domain and an intracellular SH2 phosphorylated tyrosine binding domain, a combination not found in any other organism. Tyrosine kinase signaling networks greatly expanded in the Urchoanozoan (*Manning et al., 2008*; *Pincus et al., 2008*), and the Protocadherin domain in *C. perplexa* may have been co-opted for tyrosine kinase signaling function.

Integrins are thought to have been present in the Urchoanozoan, since homologs of both components of the animal integrin plasma membrane α/β heterodimer (*Hynes, 2002*) have previously been identified in *C. owczarzaki* (*Sebé-Pedrós et al., 2010*). We found 'strong' evidence for integrin β (in the form of the diagnostic Integrin_beta Pfam domain) in only a single species of choanoflagellate, *D. costata* (*Figure 2—figure supplement 6*). In contrast, we detected the integrin β binding protein ICAP-1 (*Chang et al., 1997*), which has been proposed to act as a competitive inhibitor (*Bouvard et al., 2003*), in all choanoflagellates except *M. brevicollis* and *S. rosetta*.

## Acknowledgements

We would like to thank Monika Abedin, Rosie Alegado, Hartmut Arndt, Roman Barbalat, Greg Barton, Cédric Berney, David Booth, Lauren Booth, Bastien Boussau, Candace Britton, Thibaut Brunet, Pawel Burkhardt, Minyong Chung, Matthew Davis, Mark Dayel, Stephen Fairclough, Xavier Grau-Bové, Nicolas Henry, Ella Ireland, Tommy Kaplan, Barry Leadbeater, Tera Levin, Matthew MacManes, Kent McDonald, Alan Marron, Scott Nichols, Frank Nitsche, Ryan Null, Mathilde Paris, Iñaki Ruiz-Trillo, Devin Scannell, Monika Sigg, Jason Stajich, Alberto Stolfi, Russell Vance, Jacqueline Villalta,

Holli Weld, Jody Westbrook, Melanie Worley, Arielle Woznica, and Susan Young for assistance with laboratory techniques, bioinformatic and statistical analysis methods, helpful discussions, and comments on the manuscript. This work received a grant of computer time from the DoD High Performance Computing Modernization Program at the ERDC DoD Supercomputing Resource Center. We thank the Analytical Lab at the Marine Science Institute of the University of California, Santa Barbara for performing nutrient analysis on choanoflagellate growth media. We are grateful to the Anheuser-Busch Coastal Research Center of the University of Virginia for hosting us during collection trips for choanoflagellate isolation. This study used the Vincent J Coates Genomics Sequencing Laboratory at the University of California, Berkeley, supported by NIH S10 Instrumentation Grants S10RR029668 and S10RR027303.

## Additional information

### Funding

| Funder | Grant reference number | Author |
|---|---|---|
| Howard Hughes Medical Institute | | Michael B Eisen<br>Nicole King |
| National Institutes of Health | | Nicole King |
| U.S. Department of Defense | National Defense Science and Engineering Graduate Fellowship | Daniel J Richter |
| National Science Foundation | Central Europe Summer Research Institute Fellowship | Daniel J Richter |
| Chang-Lin Tien Fellowship in Environmental Sciences and Biodiversity | | Daniel J Richter |
| Conseil Régional de Bretagne | Postdoctoral Fellowship | Daniel J Richter |
| French Government "Investissements d'Avenir" | OCEANOMICS (ANR-11-BTBR-0008) | Daniel J Richter |
| National Science Foundation | EDEN IOS 0955517 | Parinaz Fozouni |

The funders had no role in study design, data collection and interpretation, or the decision to submit the work for publication.

### Author contributions

Daniel J Richter, Conceptualization, Data curation, Formal analysis, Investigation, Visualization, Methodology, Writing—original draft, Writing—review and editing; Parinaz Fozouni, Formal analysis, Investigation, Visualization, Methodology, Writing—review and editing; Michael B Eisen, Conceptualization, Formal analysis, Supervision, Writing—review and editing; Nicole King, Conceptualization, Formal analysis, Supervision, Funding acquisition, Writing—original draft, Project administration, Writing—review and editing

### Author ORCIDs

Daniel J Richter http://orcid.org/0000-0002-9238-5571
Michael B Eisen https://orcid.org/0000-0002-7528-738X
Nicole King http://orcid.org/0000-0002-6409-1111

### Decision letter and Author response

Decision letter https://doi.org/10.7554/eLife.34226.051
Author response https://doi.org/10.7554/eLife.34226.052

# Additional files

## Supplementary files

• Supplementary file 1. Information on each culture sequenced in this study, divided into sections by topic. See Materials and methods for details on each topic. Note that we sequenced and assembled two strains of *Stephanoeca diplocostata*. We determined their protein catalogs to be equivalent for the purposes of gene family construction, and so we used only one of the two cultures to represent the species (7.2, ATCC 50456 isolated from France). Strain information: information about the cultures used, including American Type Culture Collection (ATCC) number for each culture and NCBI Taxonomy ID for each species. Previous names for cultures indicate prior names used as labels in culture collections or publications, but which were subsequently determined to have been incorrectly applied [some of these species names are no longer valid (*Codosiga gracilis*, *Acanthoecopsis unguiculata*, *Diplotheca costata*, *Savillea micropora*, *Monosiga gracilis* and *Monosiga ovata*), whereas others are still valid at the time of publication, but their descriptions did not match the cultures (*Salpingoeca amphoridium*, *Salpingoeca minuta*, *Salpingoeca pyxidium*, *Salpingoeca gracilis* and *Salpingoeca napiformis*)]. Previous names for species were never applied to the cultures used, but are considered to be invalid names that were previously used for the species and have been subsequently replaced. Origin of cultures: source, isolation location and year, if known. Growth media, antibiotic treatments and cell types: the sequence of antibiotic treatments used to obtain a culture for high-volume growth, the growth medium and temperature, and the estimated proportion of each cell type in sequenced cultures. Culture conditions in large batches for harvesting: information on how each culture was grown and harvested for mRNA sequencing. Amount of total RNA used for each sample and groupings for preparation and sequencing: the amount of total RNA used for each culture, based on the estimated proportion of choanoflagellate RNA versus bacterial RNA in each total RNA extraction, with the goal of beginning each sequencing preparation with 2 µg of choanoflagellate total RNA. Library prep. group and Seq. flow cell group are arbitrary labels to indicate which sets of samples were prepared for sequencing at the same time or sequenced on the same flow cell. Read counts and quality trimming: results of sequencing before quality trimming and after quality trimming. Read error correction: parameters for error correction by Reptile, adapted to each set of reads independently. Assembly read counts and N50s: the counts of contigs and predicted proteins at each step of the assembly process. N50 is the length at which 50% of the nucleotides/amino acids in the assembly are contained in contigs/proteins greater than or equal to that length. NCBI Short Read Archive: identifiers to retrieve the raw, unprocessed reads for each library at the NCBI Short Read Archive. NCBI Transcriptome Shotgun Assembly (TSA): identifiers to retrieve the unannotated assembled contigs for each species, the counts of contigs entirely excluded or trimmed due to adapter or bacterial sequences identified during the submission process, and the counts of contigs removed because they were below the minimum length of 200 bases (after trimming) imposed by the TSA.

DOI: https://doi.org/10.7554/eLife.34226.029

• Supplementary file 2. Results of running BUSCO to search for conserved eukaryotic genes in each species' protein catalog. Each value represents a percentage of genes in the BUSCO eukaryotic gene set (eukaryota_odb9).

DOI: https://doi.org/10.7554/eLife.34226.030

• Supplementary file 3. Species used for gene family construction and their data sources. Note that we sequenced and assembled two strains of *Stephanoeca diplocostata*. We determined their protein catalogs to be equivalent for the purposes of gene family construction, and so we used only one of the two cultures to represent the species (ATCC 50456 isolated from France).

DOI: https://doi.org/10.7554/eLife.34226.031

• Supplementary file 4. Parsimony-based rules used to determine the presence of gene families in last common ancestors of interest and whether they represent gains on that stem lineage. Gene family presences in the five major groups (Outgroups, Fungi, Filasterea (represented by *Capsaspora owczarzaki*), Choanoflagellates and Animals) are each based on the 10% average probability threshold, as described in the Materials and methods. The same rules are presented in two alternative formats. (a) Condensed explanation, describing the criteria for presence and gain in each last common ancestor. (b) Expanded explanation, in which a 0 represents absence and a one represents presence

according to the 10% average probability threshold. Empty cells represent 0 and are omitted for clarity.

DOI: https://doi.org/10.7554/eLife.34226.032

• Supplementary file 5. Counts of gene family presence, gain and loss in last common ancestors of interest. Gains and losses are not shown for the Ureukaryote, as our data set only contained eukaryotic species and was thus not appropriate to quantify changes occurring on the eukaryotic stem lineage.

DOI: https://doi.org/10.7554/eLife.34226.033

• Supplementary file 6. Evidence levels used to determine the presence of gene families of interest. Evidence takes the form either of a protein domain architecture, the presence of a representative protein in a gene family, or a combination of the two. Protein domain names are from Pfam (transmembrane domains are from Phobius). When a domain is listed without a suffix consisting of an underscore followed by a digit, then any possible digit is acceptable (e.g., if LRR is listed, LRR_1, LRR_2, LRR_3, etc. are all acceptable). Commas indicate domain combinations, in order. Brackets are used to group combinations in order to avoid ambiguity when multiple possibilities constitute evidence for presence. Gene family IDs are from our OrthoMCL analysis and when listed, presence of a protein within that gene family is considered evidence (with a brief description of the gene family given in parentheses). *: we did not detect a canonical Toll-like receptor in *Nematostella vectensis* based on our analysis of Pfam domains in the predicted proteins from the genome assembly, but the presence of a canonical TLR has been reported in two other analyses of based on different data sets (*Miller et al., 2007*; *Sullivan et al., 2007*). **: we did not detect NF-κB in *Capsaspora owczarzaki* based on our analysis of Pfam domains in the predicted proteins from the genome assembly, but it has been previously reported to be present (*Sebé-Pedrós et al., 2011*).

DOI: https://doi.org/10.7554/eLife.34226.034

• Supplementary file 7. List of the 75 choanoflagellate-specific gene families that are present in all choanoflagellates in this study. Each feature (Panther, Pfam, Transmembrane, Signal Peptide, GO terms) is preceded by the fraction of proteins in the gene family annotated with the feature. Multiple annotations are separated by semicolons. The column 'Pfam Kinase/Phosphatase/SH2/SH3' indicates gene families that contain Pfam domains related to kinase signaling; the value shown represents the kinase-related Pfam domain with the maximum fraction annotated within the gene family.

DOI: https://doi.org/10.7554/eLife.34226.035

• Supplementary file 8. Results of a MAPLE analysis comparing the gene family content of the Urchoanozoan to the Urmetazoan, to determine gains and losses on the animal stem lineage. Only those differing in completeness by at least 25% are shown. Classifications and names are from MAPLE.

DOI: https://doi.org/10.7554/eLife.34226.036

• Supplementary file 9. Results of a MAPLE analysis comparing the gene family content of the Urchoanozoan to the Urchoanoflagellate, to determine gains and losses on the choanoflagellate stem lineage. Only those differing in completeness by at least 25% are shown. Classifications and names are from MAPLE.

DOI: https://doi.org/10.7554/eLife.34226.037

• Supplementary file 10. Results of a MAPLE analysis comparing the gene family content of the Urholozoan to the Urchoanozoan, to determine gains and losses on the choanozoan stem lineage. Only those differing in completeness by at least 25% are shown. Classifications and names are from MAPLE.

DOI: https://doi.org/10.7554/eLife.34226.038

• Supplementary file 11. Details on antibiotics tested to reduce bacterial diversity and abundance in different choanoflagellate cultures.

DOI: https://doi.org/10.7554/eLife.34226.039

• Supplementary file 12. Characteristics of growth media used. n.d.: not determined.

DOI: https://doi.org/10.7554/eLife.34226.040

• Supplementary file 13. Counts of gene family presence, gain and loss in last common ancestors of interest, as calculated by different ancestral reconstruction methods, and from a previous data set (*Fairclough et al., 2013*). Gains and losses are not shown for the last common ancestor of eukaryotes, as our data set only contained eukaryotic species and was thus not appropriate to quantify

changes occurring on the eukaryotic stem lineage. Similarly, because the Bayesian analysis (MrBayes) required an outgroup species to be specified within the data set, we could not estimate gene family presences for the last common ancestor of eukaryotes. The data set of *Fairclough et al. (2013)* included only two choanoflagellates, *M. brevicollis* and *S. rosetta*, and did not include any non-choanozoan species within Holozoa. The higher gene family counts produced by Dollo parsimony on our data set in comparison to *Fairclough et al. (2013)* were due to a combination of true and false positives: the additional species included in our analysis allowed OrthoMCL to identify a larger set of truly orthologous genes, but false positives also resulted from providing many additional species' protein catalogs as input to OrthoMCL without the subsequent application of a probability-based approach to filter out spurious BLAST hits.

DOI: https://doi.org/10.7554/eLife.34226.041

• Supplementary file 14. Representative signaling genes in the animal TLR-NFκB pathway with their representatives in mouse (*Mus musculus*). Functions and roles are derived from the UniProt database. Pfam domain architectures are separated by tildes (~) and are listed in order. When two architectures are possible, they are separated by 'or'. Pfam domain architecture phylogenetic distribution represents the phylogenetic group containing all of the species with that architecture in the Pfam database. Similarly, gene family phylogenetic distribution represents the phylogenetic group containing all of the species with representative proteins in the gene family. Notes contains either a description of the species/genes present in the gene family, or a brief summary of the blastp hits in the NCBI nr database for proteins encoded by select species in the gene family, which we used to verify the assignment of the protein to the gene family.

DOI: https://doi.org/10.7554/eLife.34226.042

• Transparent reporting form

DOI: https://doi.org/10.7554/eLife.34226.043

## Data availability

Raw sequencing reads have been deposited at the NCBI SRA under BioProject PRJNA419411 (19 choanoflagellate transcriptomes) and PRJNA420352 (S. rosetta polyA selection test). Transcriptome assemblies, annotations, and gene families are available on figshare at DOI: 10.6084/m9.figshare.5686984.v2. Transcriptome assemblies have also been submitted to the NCBI Transcriptome Shotgun Assembly database under BioProject PRJNA419411. Protocols have been deposited to protocols.io and are accessible at DOI: 10.17504/protocols.io.kwscxee.

The following datasets were generated:

| Author(s) | Year | Dataset title | Dataset URL | Database, license, and accessibility information |
|---|---|---|---|---|
| Richter DJ, Fozouni P, Eisen MB, King N | 2017 | Transcriptome sequencing of 19 diverse species of choanoflagellates | https://www.ncbi.nlm.nih.gov/bioproject/PRJNA419411/ | Publicly available at NCBI BioProject (Accession no: PRJNA419411) |
| Richter DJ, Fozouni P, Eisen MB, King N | 2017 | Test of polyA selection to separate choanoflagellate from bacterial RNA | https://www.ncbi.nlm.nih.gov/bioproject/PRJNA420352/ | Publicly available at NCBI BioProject (Accession no: PRJNA420352) |
| Richter DJ, Fozouni P, Eisen MB, King N | 2017 | Data from: Gene family innovation, conservation and loss on the animal stem lineage | https://dx.doi.org/10.6084/m9.figshare.5686984.v2 | Available at figshare under a Creative Commons Attribution 4.0 (CC-BY) license |

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
