## [Decision Letter]

Thank you for submitting your article "The ancestral animal genetic toolkit revealed by diverse choanoflagellate transcriptomes" for consideration by *eLife*. Your article has been reviewed by three peer reviewers, and the evaluation has been overseen by a Reviewing Editor and Patricia Wittkopp as the Senior Editor. The following individuals involved in review of your submission have agreed to reveal their identity: Casey Dunn (Reviewer #2); Warren Francis (Reviewer #3).

The reviewers have discussed the reviews with one another and the Reviewing Editor has drafted this decision to help you prepare a revised submission.

Summary:

This is an extremely interesting and very well executed paper that sets a new benchmark in understanding the evolution of gene content within and between animals, choanoflagellates, and their relatives. This is fundamental to answering basic questions about the history of the evolution of genomes and many phenotypes of broad interest.

The authors have sequenced 19 new choanoflagellate transcriptomes and used them to reassess which animal genes have pre-metazoan origins. It's an interesting paper – the data are valuable.

- The results are biologically interesting. It clearly pinpoints in much greater detail than previous studies the history of gene gain and loss along key branches in the tree of life.

- It will be a great technical resource. Investigators interests in particular gene families can refer to this paper to understand where their favorite genes have been gained and lost, and which organisms they are in. Investigators interested in particular branches in this portion of the tree can refer to the paper to see how gene inventory changed along these branches.

I found the results to be well presented, and appreciated how clear and clean the figures are.

The paper does much to clarify what doesn't make animals special – an essential part of understanding what does make them special. Like other lineages, animals have lost many genes. There has not been a march toward increased gene content in animals, with other extant lineages representing snapshots in animal history, as is often implied or assumed (based on scant evidence). There has been gain and loss of genes along many branches, including a similar rate of gain and loss in animals and choanoflagellates.

Essential revisions:

Phylogeny: Please at minimum discuss the following in any response. If necessary comment on the possible impact of using a different phylogeny.

The authors reference the phylogeny of Carr et al., 2017 for the interpretation of their results (e.g. Figure 3). This analysis is based on 6 genes, and differs somewhat from a phylogenomic analysis in Simion et al.. 2017 e.g. in the position of *Codosiga* and *Salpingoeca dolichothecata*. I think these discrepancies are important as they affect how parsimonious some of the purported choanoflagellate gene losses are. It is a shame that the authors do not present a more definitive phylogeny using the data that they have produced, or at least discuss interpretative issues in light of different phylogenetic possibilities (even if to say there are no serious issues).

Methodology: My major concern is with the OrthoMCL based approach – it really isn't clear what an OrthoMCL gene family corresponds to (in terms of orthologs/paralogs). This makes summary numbers suspect (1700 gene families with origins in the animal stem lineage of which 36 are conserved across animals). What is an OrthoMCL gene family in terms of true biological entities?

The authors highlight a number of specific examples of gene loss from *M. brevicollis* and *S. rosetta*, (TLRs, Delta etc.), but leaving these vignettes aside, and the 350 number in the Abstract (which I don't find in the main text), I cannot find a discussion of how anomalous/representative the authors consider the two taxa with sequenced genomes to be. It would be helpful to include this.

Table 1. I think there's a difference between what the reader is likely to think has been done and what actually has been done, and as the 36 number makes it to the Abstract, it's quite important to be clear. WNT ligands, for instance are a classic example of a metazoan novelty (and to the best of my knowledge have still not been reported in non-animals). The OrthoMCL based protocol breaks up these ligands into a number of different animal specific families, none of which feature in Table 1, perhaps because no single family is present in all animals studied. Yet the innovation itself, the invention in the animal stem lineage of the first Wnt seems to me as though it is exactly the sort of thing the paper is suggesting it will report. I would like to see a clearer discussion of what the 36 number means and how robust it is to methodological approach. If I constructed an alignment and phylogeny for each gene in the dataset and assessed when it had an ortholog in all other animals would I get a number even close to 36? Would the authors be happy for people to cite this paper saying that there were only 36 conserved animal specific gene families?

Surprised that no gene trees are presented. Phylogenetic analyses of gene families could do much to clarify some of the patterns that are discussed. For example, in the section "Construction of gene families and their probabilities of presence" the authors hypothesize that cases where a gene is present in one choano but many animals are due to false positives with blast. Building a multiple sequence alignment and gene tree would be a simple and informative way to evaluate this. There is also the risk that some choano data could be contaminated with animal sequences, and phylogenetic trees would help clarify this.

It also isn't clear to me why the authors didn't use standard maximum likelihood and Bayesian character evolution methods to trace the gain and loss of gene families on the species phylogeny. This would also infer ancestral state probabilities for gene presence/absence under explicit models of evolution. Such methods are well established and highly relevant here.

Subsection “The phylogenetic distribution of animal and choanoflagellate gene families”, first paragraph: I do not think that the use of BUSCO is appropriate for measuring the completeness of transcriptomes, particularly as the whole point of this study was to examine the gene content of choanos relative to animals, hence of a group which is not currently well-represented (for instance, it does not appear that *Monosiga* or *Salpingoeca* were in the BUSCO set).

How many gene families/proteins are recovered by the transcriptomes for either or both of the species with genomes i.e. does the RNAseq get most of the genes predicted from the genomes? Is there any reason to consider that the gene numbers vary substantially from *Monosiga* and *Salpingoeca* (roughly 10k genes)? For instance, in the Supplementary file 1, *Codosiga* has 61k proteins, but presumably fewer "genes" if these were clustered. This is mentioned in the Materials and methods, but may be better explicitly discussed in the Results.

Gene family construction, subsection “Construction of gene families and their probabilities of presence”: Relating to whether the "residual probability" genes are true orthologs, it would appear that this method might systematically miss gene presence from taxa with long-branches. One can consider cases where true orthologs have good matches within groups of related species (say within flatworms or roundworms) but other taxa are too distant to have good matches. This was found to be the case in flatworms (see Martin-Duran et al., 2017 Genome Research). This could imply that homology of some fast evolving genes may be overlooked between animals and choanoflagellates. Perhaps discuss how many genes simply have no matches between metazoa and choanoflagellates vs. weak matches.

Subsection “Animal-specific gene families: innovation and loss”, first paragraph: GO terms are often controversial (too broad or wrong) for non-model organisms, a problem highlighted on the GO website. The GO terms discussed in the text are plausible, but the supplemental table also contains a number of implausible ones, such as heterocycle biosynthesis or aromatic compounds biosynthesis, which have smaller p-values than the ones discussed. It is unclear if animals have these pathways as the terms might be carryover from regulatory elements, but the presence of these pathways is also contradicted later saying they do not have the shikimate pathway, etc. Because any annotations are biased by model animals, very little is known about possible functions in choanoflagellates or non-model animals, making it difficult to say any function in early animals (i.e. Burkhardt et al., 2014 Mol Biol Evo). It may be too drastic to say these results should be removed entirely, but perhaps very critically discussed, esp. in light of the results on Figure 2—figure supplement 7, suggesting most genes have no annotation.

ECM Experiments: The referees agree that the final section on ECM experiments does not sit well with the rest of the manuscript. It also does not add a great deal and there is insufficient space to do the experiments justice. We feel this should be omitted from a revision.

---

## [Author Response]

Essential revisions:Phylogeny: Please at minimum discuss the following in any response. If necessary comment on the possible impact of using a different phylogeny…..The authors reference the phylogeny of Carr et al., 2017 for the interpretation of their results (e.g. Figure 3). This analysis is based on 6 genes, and differs somewhat from a phylogenomic analysis in Simion et al.. 2017 e.g. in the position of Codosiga and Salpingoeca dolichothecata. I think these discrepancies are important as they affect how parsimonious some of the purported choanoflagellate gene losses are. It is a shame that the authors do not present a more definitive phylogeny using the data that they have produced, or at least discuss interpretative issues in light of different phylogenetic possibilities (even if to say there are no serious issues).

We agree with the reviewer’s comment that the different tree topologies among published choanoflagellate phylogenies had the potential to lead to different inferences about choanoflagellate gene loss. Knowing this, we designed our principal analyses of choanoflagellate gene loss to be largely independent of the internal topology of the choanoflagellate phylogeny. We addressed the issues surrounding choanoflagellate phylogenetics in three specific ways:

1) Early in our analyses of choanoflagellate transcriptomes, we performed phylogenetic analyses using protein sequences from hundreds of gene families, with the goal of generating a robust choanoflagellate phylogeny. We found that the positions of the two species highlighted by the reviewers (*Codosiga hollandica* and *Salpingoeca dolichothecata)* were inconsistent among different gene family sets and phylogenetic inference methods, and that their branches frequently had low support values. We believe this is because both species are isolated on long terminal branches. Genome-scale sequence data are currently unavailable from closely-related sister groups to either species (e.g., other *Codosiga* species for *C. hollandica,* or *Salpingoeca tuba* for *S. dolichothecata*). Moreover, we considered the construction of a robust species tree to be outside the scope of the present work. Accordingly, we added the following sentence to the Results: “Although the focus of our study was on reconstructing large-scale patterns of gene family evolution between animals and choanoflagellates, and not on phylogenetics, the availability of a comprehensive set of gene predictions raised the possibility that we could generate a well-supported and well-resolved phylogeny of choanoflagellates. However, we found that the positions of two species lying on long terminal branches (*Salpingoeca dolichothecata* and *Codosiga hollandica*) were poorly supported or recovered at inconsistent locations.” We also updated the Materials and methods (subsection “Species tree and phylogenetic diversity”) with details of these analysis.

2) Importantly, we designed the principal analysis of our paper, the reconstruction of ancestral gene content, to be independent of the presently unresolved phylogenetic relationships within choanoflagellates or within animals. To accomplish this, for gene families shared among major groups of species (e.g., choanoflagellates, animals, fungi), we applied a 10% minimum average probability criterion (Materials and methods, subsection “Group-specific core gene families found in all extant members”, first paragraph) for gene family presence that did not depend on the phylogeny within the group. Similarly, for gene families found only in choanoflagellates or in animals, to be considered present in the common ancestor of the group, we required at least two sub-groups of species to pass the 10% average probability criterion. We chose sub-groups such that our results should not depend on currently unresolved phylogenetic relationships. For choanoflagellates, the sub-groups were craspedids and loricates (which are well-defined groups (Leadbeater, 2015; Carr et al., 2017); see Supplementary file 4). To emphasize this aspect of our analysis, we added the following sentence to the Results: “Therefore, to avoid basing our comparative genomics efforts on a potentially incorrect phylogeny, we designed our analyses to be independent of species relationships within either animals or choanoflagellates. [For display purposes only, we relied on a consensus of previously published phylogenies (Philippe et al., 2009; Burki et al., 2016; Carr et al., 2017)].” We also added further details in the Materials and methods (subsection “Species tree and phylogenetic diversity”, second paragraph).

3) We agree with the reviewer that the parsimony of gene losses within choanoflagellates necessarily depends on the topology of their species tree. Because of current uncertainties in the phylogenetic relationships among choanoflagellates, and the fact that this study is most focused on reconstructing the origin of animal genomes, we deferred analyses of gene family loss within choanoflagellates to future studies, and we discussed gene family loss on the choanoflagellate stem lineage as an aside in the Results: “[Although these amino acid synthesis and osmosensing pathway components were retained in choanoflagellates, several other gene families involved in diverse biosynthetic pathways were instead lost on the choanoflagellate stem lineage (Supplementary file 9).]” We also note that the gene family loss we observed was frequently so widespread within choanoflagellates that numerous independent losses would be required regardless of the species tree topology (e.g., RNAi machinery, Materials and methods subsection “RNAi machinery in choanoflagellates” and Figure 3—figure supplement 2). This is also true within the animals (see, for example, Figure 3 or Figure 3—figure supplement 3).

Methodology: My major concern is with the OrthoMCL based approach – it really isn't clear what an OrthoMCL gene family corresponds to (in terms of orthologs/paralogs). This makes summary numbers suspect (1700 gene families with origins in the animal stem lineage of which 36 are conserved across animals). What is an OrthoMCL gene family in terms of true biological entities?

The goal of the OrthoMCL approach was to identify true orthologous gene families. There are currently no optimal strategies to achieve this goal for genome-scale analyses (Altenhoff et al., 2016), and we therefore chose OrthoMCL because it provides a reasonable balance of sensitivity and specificity compared to other methods (Altenhoff and Dessimoz, 2009). Furthermore, we coupled OrthoMCL with a technique to remove non-orthologous sequences from resulting gene families (described in the Materials and methods, subsection “Construction of gene families and their probabilities of presence”). We concluded that this technique was effective in enriching gene families for true orthologs (Figure 2—figure supplement 2f); please also see our response to the comment on phylogenetic trees below, in which we tested the technique with a phylogenetic approach.

To address the reviewer’s concern, when we first introduce OrthoMCL gene families (Results), we now refer the reader to a discussion in the Materials and methods on their interpretation, as follows: “We next compared the choanoflagellate gene catalogs with those of diverse animals and phylogenetically relevant outgroups (Supplementary file 3) to identify orthologous gene families and determine the ancestry of genes present in animals (see Materials and methods for rationale underlying inferences of gene family orthology).” We updated and clarified the two corresponding sections in the Materials and methods: “Our goal in constructing gene families was to identify groups composed of orthologous proteins. Although numerous approaches are currently available, no existing algorithm can yet identify all orthologous genes while perfectly separating orthologs from spurious BLAST hits (Altenhoff et al., 2016). We chose OrthoMCL due to its widespread use in previous studies, including analyses of animal gene family evolution based on the genomes of *S. rosetta* and *M. brevicollis* (Fairclough et al., 2013), and due to its relative balance of sensitivity and specificity in comparison to other algorithms (Altenhoff and Dessimoz, 2009)” and “We conclude that this approach was able to identify and remove a substantial proportion of false orthologs, resulting in a set of gene families highly enriched for (although still not entirely composed of) truly orthologous families.”

In addition, throughout our manuscript, in order to emphasize the uncertainties in using OrthoMCL gene families, we precede all estimated gene family counts with a tilde (~). Furthermore, we focus on relative comparisons of gene family counts rather than on the absolute number of gene families lost or gained; for example, in the Results: “the numbers of gene families gained on the animal and choanoflagellate stem lineages are roughly equivalent (~1,944 and ~2,463, respectively), indicating that the specific functions of novel gene families, not their quantity, were critical to the very different phenotypes each clade went on to have.” This is further discussed in the Materials and methods.

The authors highlight a number of specific examples of gene loss from M. brevicollis and S. rosetta, (TLRs, Delta etc.), but leaving these vignettes aside, and the 350 number in the Abstract (which I don't find in the main text), I cannot find a discussion of how anomalous/representative the authors consider the two taxa with sequenced genomes to be. It would be helpful to include this.

In our analysis, any gene family present in multiple choanoflagellate species would be inferred to have been present in their last common ancestor (see Materials and methods subsection ““Group-specific core gene families found in all extant members”). Therefore, we were able to determine how representative *M. brevicollis* and *S. rosetta* are compared to other choanoflagellate species by calculating the number of gene families they retained from the last common ancestor of choanoflagellates, and subsequently comparing their gene retention to other choanoflagellates, as presented in Figure 3 (in particular, Figure 3c) and its figure supplements. To clarify the relevance of this analysis in responding to the reviewer’s question, we updated the Results, as follows: “Notably, the two choanoflagellate species with previously-sequenced genomes, *M. brevicollis* and *S. rosetta*, retain among the fewest ancestral gene families, indicating that they are less representative of Urchoanoflagellate gene content than are most choanoflagellate species we sequenced.”

The 350 number in the Abstract that was difficult to find in the main text referred to the number of gene families previously thought to have been animal-specific that are now found in choanoflagellates. To respond to this point (and to another reviewer suggestion that numbers in the Abstract correspond exactly to those in the text) we have revised this value to ~372 in the Abstract so that it corresponds directly to the value discussed in the Introduction and the Results.

Table 1. I think there's a difference between what the reader is likely to think has been done and what actually has been done, and as the 36 number makes it to the Abstract, it's quite important to be clear. WNT ligands, for instance are a classic example of a metazoan novelty (and to the best of my knowledge have still not been reported in non-animals). The OrthoMCL based protocol breaks up these ligands into a number of different animal specific families, none of which feature in Table 1, perhaps because no single family is present in all animals studied. Yet the innovation itself, the invention in the animal stem lineage of the first Wnt seems to me as though it is exactly the sort of thing the paper is suggesting it will report. I would like to see a clearer discussion of what the 36 number means and how robust it is to methodological approach. If I constructed an alignment and phylogeny for each gene in the dataset and assessed when it had an ortholog in all other animals would I get a number even close to 36? Would the authors be happy for people to cite this paper saying that there were only 36 conserved animal specific gene families?

Thank you for raising this important point. We have tried to clarify the meaning of the core animal-specific gene families, while also expanding our analysis, as follows:

1) We have added the following sentence to the Results immediately after introducing core animal-specific gene families: “This count of core animal-specific gene families is likely to be an underestimate due to methodological tradeoffs in the genome-scale analysis that we used to identify gene families (see Materials and methods).” We also modified the Materials and methods to add the scenario proposed by the reviewer (orthologous genes broken up into a number of different animal-specific gene families) to the list of potential technical artifacts that was already present, which now reads as follows: “These core gene families are subject to several potential technical artifacts. First, an incomplete genome or transcriptome assembly could result in a species appearing to lack a gene family. […] Thus, the lists of core gene families should not be considered exhaustive, especially for serially duplicated or repeat-containing gene families.”

2) We added an additional analysis to make it clear to readers that, although there are very few core animal-specific gene families, there are many other animal-specific gene families missing in only a small number of animals. This analysis is reflected in a new figure and in a new sentence in the Results: “By reducing the stringency of the requirement for conservation, we identified a total of 153 gene families that were missing in no more than two animals from our data set (i.e., approximately 10%; Figure 3—figure supplement 3)”.

Surprised that no gene trees are presented. Phylogenetic analyses of gene families could do much to clarify some of the patterns that are discussed. For example, in the section "Construction of gene families and their probabilities of presence" the authors hypothesize that cases where a gene is present in one choano but many animals are due to false positives with blast. Building a multiple sequence alignment and gene tree would be a simple and informative way to evaluate this. There is also the risk that some choano data could be contaminated with animal sequences, and phylogenetic trees would help clarify this.

We appreciate the reviewer’s suggestion to construct phylogenetic trees in order to clarify the patterns of gene family evolution we discussed. Although this is not feasible on a genome-wide scale, we performed three separate types of phylogenetic analyses to address the concerns raised in this comment:

1) To test the ability of our technique to identify false positives in gene families (e.g., those with few choanoflagellates and many animals), we selected the representative gene family presented in Figure 2—figure supplement 2c. This gene family, 9066, contains numerous animal proteins but only a single choanoflagellate protein. The choanoflagellate protein was assigned a probability of 0.03 due to its high average blastp E value to other members of the family. We searched for closely-related outgroup sequences from our full data set and built a phylogenetic tree for the gene family including the additional outgroup sequences. The choanoflagellate protein was indeed more closely related to the outgroup sequences than to any of the animal proteins inside the gene family, and thus correctly identified as a false positive by our probability assignment method (as presented in a new figure, Figure 2—figure supplement 10). We added the details of this analysis to the Methods (subsection “Construction of gene families and their probabilities of presence”).

2) To assess whether the choanoflagellate transcriptomes we produced were contaminated with animal sequences, we built phylogenetic trees for all seven protein coding genes with previously available sequences for multiple choanoflagellates in GenBank (which we used as a positive control to verify species identity). We observed no instances with ≥ 50% bootstrap support in which a choanoflagellate sequence was nested within a clade of animal sequences, indicating that none of the choanoflagellate transcriptome sequences we tested appeared to be of animal origin (newly produced tree files are in Figure 2—source data 1). These results are detailed in a new section of the Materials and methods, “Tests of choanoflagellate species identity and contamination with animal sequences”. They also confirm the effectiveness of the protocols we implemented to avoid contamination during each step of the mRNA preparation and sequencing process, as described in the Materials and methods.

3) To validate previously animal-specific gene families we newly identified as present in choanoflagellates, we built phylogenetic trees for three examples: TLRs (Figure 4—figure supplement 2), Notch and Delta (Figure 2—figure supplement 12). In each case, there were no well-resolved branching patterns contradicting the hypothesis that these three gene families evolved on the stem lineage leading to the last common ancestor of animals and choanoflagellates. We added details for these analyses to the Materials and methods (TLRs, subsection “Toll-like receptor signaling and innate immunity”; Notch and Delta, subsection “Notch and Delta”).

It also isn't clear to me why the authors didn't use standard maximum likelihood and Bayesian character evolution methods to trace the gain and loss of gene families on the species phylogeny. This would also infer ancestral state probabilities for gene presence/absence under explicit models of evolution. Such methods are well established and highly relevant here.

A novel aspect of our analysis was the calculation of probabilities of presence for each species within each gene family generated by OrthoMCL. We demonstrated that probabilities are more appropriate than simple presence or absence (see Figure 2—figure supplement 2, Materials and methods subsection “Construction of gene families and their probabilities of presence”, and above in our response). We are not aware of any existing method for phylogenetic reconstruction of character evolution that is capable of accepting probabilities as inputs (e.g., the 24 ancestral reconstruction methods surveyed in (Joy et al., 2016)). Therefore, we defined an average probability threshold for ancestral reconstruction. Instead of using simple presence/absence, the average threshold retains the uncertainty represented by individual probabilities within a group of species (animals, choanoflagellates, etc.) to determine whether a gene family was present or absent within the group (described in Materials and methods, subsection “Test of existing methods of ancestral gene content reconstruction”).

To implement the reviewer’s suggestion, we also tested three types of existing ancestral reconstruction methods using presence/absence data as input: Dollo parsimony, maximum likelihood, and Bayesian. We found that these methods each produced substantially different estimates of ancestral gene family content. For example, for the last common ancestor of animals and choanoflagellates, estimates ranged from 6,679 gene families present (maximum likelihood via Mesquite) to 19,579 (Dollo parsimony via PHYLIP) to 29,519 (Bayesian via MrBayes); complete results are in a new table, Supplementary file 13. Full details of these analyses can be found in a new section of the Methods, “Test of existing methods of ancestral gene content reconstruction”.

Thus, we feel our technique represents an improvement over existing methods due to its ability to input probabilities for gene family presence, while we eagerly await the development of maximum likelihood or Bayesian methods capable of doing the same. The estimates of gene family innovation based on our technique were also congruent with previous analyses conducted before our data set became available (Srivastava et al., 2010; Fairclough et al., 2013; Simakov and Kawashima, 2017). In recognition of the uncertainties in ancestral gene content reconstruction, as described in our response above, throughout our manuscript we precede all estimated gene family counts with a tilde (~) and we focus on relative comparisons of gene family counts rather than on the absolute number of gene families lost, present or gained.

Subsection “The phylogenetic distribution of animal and choanoflagellate gene families”, first paragraph: I do not think that the use of BUSCO is appropriate for measuring the completeness of transcriptomes, particularly as the whole point of this study was to examine the gene content of choanos relative to animals, hence of a group which is not currently well-represented (for instance, it does not appear that Monosiga or Salpingoeca were in the BUSCO set).How many gene families/proteins are recovered by the transcriptomes for either or both of the species with genomes i.e. does the RNAseq get most of the genes predicted from the genomes? Is there any reason to consider that the gene numbers vary substantially from Monosiga and Salpingoeca (roughly 10k genes)? For instance, in the Supplementary file 1, Codosiga has 61k proteins, but presumably fewer "genes" if these were clustered. This is mentioned in the Materials and methods, but may be better explicitly discussed in the Results.

The purpose of the comparison using BUSCO was to demonstrate that none of the new transcriptomes was significantly deficient in terms of gene content, and thus suitable for further analysis. This was the case, since the conserved gene content of the transcriptomes we produced was similar to the two sequenced choanoflagellate genomes. Regarding recovery of gene families by transcriptomes, the analysis we presented in Figure 1—figure supplement 2a estimated the proportion of genes predicted from the *Salpingoeca rosetta* genome that were present in its transcriptome (as part of a test of protocols we modified for polyA selection; Materials and methods, subsection “Test of polyA selection to separate choanoflagellate from bacterial RNA”). This analysis indicated that 93% of genes predicted from the genome (green line) were represented in the transcriptome over ≥ 90% of their length (yellow line). We have now tried to present this analysis more clearly.

In response to the reviewer’s comment, we reorganized the corresponding section in the Results, first to report the completeness of the *S. rosetta* transcriptome versus the genome, and next to present BUSCO results, as follows: “Using multiple independent metrics, we found that the new choanoflagellate transcriptomes approximate the completeness of choanoflagellate genomes for the purposes of cataloging protein-coding genes. […] Furthermore, compared with the genomes of *M. brevicollis* and *S. rosetta*, which contain 83% and 89%, respectively, of a benchmark set of conserved eukaryotic genes [BUSCO; (Simão et al., 2015)], each of the new choanoflagellate transcriptomes contains between 88 – 96% (Supplementary file 2).” We also updated the legend of Figure 1—figure supplement 2a and the Materials and methods.

To address the reviewer’s comment regarding the variation in predicted proteins from the new choanoflagellate transcriptomes, we added the following sentences to the Results: “After performing de novo transcriptome assembly and filtering for cross-contamination, we predicted a catalog of between 18,816 – 61,053 unique protein-coding sequences per species. [These counts likely overestimate the true numbers of underlying protein-coding genes, as they may include multiple alternative splice variants for any given gene and redundant contigs resulting from intra-species polymorphisms or sequencing artifacts (Grabherr et al., 2011; Haas et al., 2013).]”

Gene family construction, subsection “Construction of gene families and their probabilities of presence”: Relating to whether the "residual probability" genes are true orthologs, it would appear that this method might systematically miss gene presence from taxa with long-branches. One can consider cases where true orthologs have good matches within groups of related species (say within flatworms or roundworms) but other taxa are too distant to have good matches. This was found to be the case in flatworms (see Martin-Duran et al., 2017 Genome Research). This could imply that homology of some fast evolving genes may be overlooked between animals and choanoflagellates. Perhaps discuss how many genes simply have no matches between metazoa and choanoflagellates vs. weak matches.

Two aspects of our methods address fast-evolving gene families:

1) We designed the calculation of probabilities from BLAST E values to be permissive enough to allow the possibility of weak homology (while trying to balance sensitivity versus specificity for spurious hits).

2) We chose an average probability threshold of 10% such that a relatively small number of species with high average probability would be sufficient for the gene family to be present within a group.

In response to the reviewer’s comment, we rewrote the section of the Materials and methods that discusses the 10% probability threshold: “For choanoflagellates and animals, each of which have 21 species in our data set, this equates, for example, to a gene family represented at high probability (≥ 70%, corresponding to an average BLAST E value of 1 x 10^-20^; Figure 2—figure supplement 2a) in three or more species, or at 35% probability (E value of 1 x 10^-10^) in six or more species. Therefore, the only truly orthologous gene families likely to be excluded by this threshold are those with weak homology (i.e., average BLAST E values > 10^-10^) to only a few species in a major group, a rare case which would also require numerous independent losses of the gene family within the group.”

In addition, we modified Figure 1—figure supplement 2b to indicate the correspondence between a probability of 10% and an individual BLAST E value of between 10^-6^ and 10^-7^.

In the example case proposed by the reviewer (gene families in which a taxon of related long-branched animals have remote homology to the rest of the gene family), they would only be considered missing in animals if they were absent (or present at very low probability) from nearly all other animals outside the long-branched taxon. This, in turn, would imply numerous independent losses of the gene family within animals. In our method, this represented a rare case necessary to exclude in order to create an effective balance between sensitivity and specificity for our global analysis.

Subsection “Animal-specific gene families: innovation and loss”, first paragraph: GO terms are often controversial (too broad or wrong) for non-model organisms, a problem highlighted on the GO website. The GO terms discussed in the text are plausible, but the supplemental table also contains a number of implausible ones, such as heterocycle biosynthesis or aromatic compounds biosynthesis, which have smaller p-values than the ones discussed. It is unclear if animals have these pathways as the terms might be carryover from regulatory elements, but the presence of these pathways is also contradicted later saying they do not have the shikimate pathway, etc. Because any annotations are biased by model animals, very little is known about possible functions in choanoflagellates or non-model animals, making it difficult to say any function in early animals (i.e. Burkhardt et al., 2014 Mol Biol Evo). It may be too drastic to say these results should be removed entirely, but perhaps very critically discussed, esp. in light of the results on Figure 2—figure supplement 7, suggesting most genes have no annotation.

We agree with the reviewer’s argument that results of the Gene Ontology analysis we performed should be interpreted with caution. While we reluctantly included them in our original analysis as a point of comparison with other studies, we were happy to remove the GO analysis from our revised manuscript.

ECM Experiments: The referees agree that the final section on ECM experiments does not sit well with the rest of the manuscript. It also does not add a great deal and there is insufficient space to do the experiments justice. We feel this should be omitted from a revision.

We agree with the reviewers and have omitted results related to ECM from the revised manuscript.